# Functional expression of inwardly rectifying and ATP-sensitive potassium channels in human pulmonary artery smooth muscle and endothelial cells

Bianca Barreira[1,2,3] 🆔, Daniel Morales-Cano[1,2,3], Laura Moreno[1,2,3], Beatriz de Olaiz[4], Rui Adão[1,2,3], Angel Cogolludo[1,2,3] 🆔, Francisco Perez-Vizcaino[1,2,3] 🆔 and Maria Sancho[2,3,5,6] 🆔

[1]*Department of Pharmacology and Toxicology, School of Medicine, Universidad Complutense de Madrid, Madrid, Spain*
[2]*Centro de Investigación Biomédica en Red de Enfermedades Respiratorias (CIBERES), Madrid, Spain*
[3]*Instituto de Investigación Sanitaria Gregorio Marañón (IiSGM), Madrid, Spain*
[4]*Department of Thoracic Surgery, Hospital Universitario de Getafe, Getafe, Spain*
[5]*Department of Physiology, School of Medicine, Universidad Complutense de Madrid, Madrid, Spain*
[6]*Department of Pharmacology, University of Vermont, Burlington, Vermont, USA*

Handling Editors: Kim Barrett & Nikki Jernigan

The peer review history is available in the Supporting information section of this article (https://doi.org/10.1113/JP289445#support-information-section).

**Abstract figure legend** Inwardly rectifying (Kir2) and ATP-sensitive ($K_{ATP}$) potassium channels are functionally expressed in human pulmonary artery endothelial and smooth muscle cells. The schematic illustrates how Kir2- and $K_{ATP}$-mediated $K^+$ efflux contributes to $V_M$ regulation and pulmonary vascular tone. Under resting conditions, Kir2 channels are tonically active, and pharmacological activation of $K_{ATP}$ channels by pinacidil promotes $V_M$

**Bianca Barreira** is a final-year PhD student in the Department of Pharmacology and Toxicology at the Medical School, Universidad Complutense de Madrid (Spain). Her research has helped unravel key physiological and pathophysiological mechanisms underlying pulmonary circulation. She currently investigates the role of Kir2 and $K_{ATP}$ channels in the control of pulmonary blood flow and their involvement in pulmonary arterial hypertension. With extensive experimental experience, she adopts an integrative research approach that links fundamental ion channel function to disease-relevant conditions, bridging basic discoveries with potential translational applications.

The Journal of Physiology

hyperpolarization and supports vasodilatation. Conversely, inhibition with micromolar $Ba^{2+}$ (Kir2 blocker) or glibenclamide/PNU-37883A ($K_{ATP}$ blockers) depolarizes the membrane and induces vasoconstriction.

**Abstract** The resting membrane potential ($V_M$) of vascular cells is a key determinant of arterial tone, integrating multiple ionic conductances to control smooth muscle contractility and endothelial signalling. In the human pulmonary circulation, the specific $K^+$ channels responsible for setting the $V_M$ of smooth muscle cells (SMCs) and endothelial cells (ECs) remain incompletely defined. This study investigated whether inwardly rectifying (Kir2) and ATP-sensitive ($K_{ATP}$) $K^+$ channels are functionally expressed in native human pulmonary artery (PA) SMCs and ECs and assessed their contribution to vascular tone. Combining patch-clamp electrophysiology, immunofluorescence and wire myography, we evaluated channel expression and function in freshly isolated PASMCs and PAECs, and intact PAs. Kir2 channels were identified by $Ba^{2+}$-sensitive inward currents with a characteristic rectification profile, supported by immunolabelling of Kir2.1 and Kir2.2 subunits. Functionally, $BaCl_2$ induced concentration-dependent contractions of PA rings and significantly attenuated acetylcholine-evoked, endothelium-dependent relaxation, revealing a tonic vasodilatory role for Kir2 channels. $K_{ATP}$ currents, activated by pinacidil and blocked by glibenclamide and PNU-37883A, were also observed in PASMCs and PAECs, consistent with immunodetection of Kir6.1 and SUR2 subunits. In isolated PAs, pinacidil elicited concentration-dependent vasodilatation, which was significantly reduced by $K_{ATP}$ channel blockade. Collectively, these findings demonstrate for the first time the functional presence of Kir2 and $K_{ATP}$ channels in native human pulmonary vascular cells, and their modulatory role on $V_M$ and arterial tone. These channels emerge as key electro-metabolic regulators of pulmonary vascular function and promising therapeutic targets in diseases characterized by $V_M$ dysregulation, such as pulmonary arterial hypertension.

(Received 10 June 2025; accepted after revision 6 January 2026; first published online 29 January 2026)

**Corresponding author** M. Sancho: Department of Physiology, Faculty of Medicine, Universidad Complutense de Madrid, Plaza Ramon y Cajal S/N, Madrid, 28040, Spain. Email: masanc75@ucm.es

**Key points**

- Inwardly rectifying (Kir2) $K^+$ channels are key regulators of the resting membrane potential ($V_M$) in different vascular cell types across multiple vascular beds, whereas ATP-sensitive ($K_{ATP}$) $K^+$ channels detect changes in the metabolic state of vascular cells and translate these changes into $V_M$ modulation.
- Despite their well-established physiological relevance, a comprehensive characterization of Kir2 and $K_{ATP}$ channels in freshly isolated human pulmonary vascular cells – particularly within the endothelium – remains lacking.
- Our study provides compelling evidence for the functional expression of Kir2 and $K_{ATP}$ channels in native human pulmonary arterial smooth muscle and endothelial cells, demonstrating their contribution to $V_M$ regulation and pulmonary vascular tone at rest and in response to specific stimuli.

## Introduction

The pulmonary circulation operates under unique regulatory mechanisms that ensure efficient gas exchange while maintaining low vascular resistance under physiological conditions (Glenny & Robertson, 2011). This finely tuned control relies on the dynamic interplay of distinct ion channels and the propagation of electrical signals through gap junctions – key determinants of the resting membrane potential ($V_M$) in vascular cells (Mironova et al., 2024; Mondéjar-Parreño et al., 2021; Welsh et al., 2018). In resistance arteries and arterioles, $V_M$ functions as a central integrator of electrical and mechanical signalling, coupling membrane excitability to contractile responses (i.e. myogenic tone) (Nelson et al., 1990; Tykocki et al., 2017). By modulating the contractility of smooth muscle cells (SMCs), $V_M$ directly regulates vessel diameter and pulmonary vascular resistance, thereby

protecting the heart from excessive afterload and aligning capillary perfusion with alveolar ventilation to optimize gas exchange (Platoshyn et al., 2000).

In vascular SMCs, the steady-state $V_M$ results from the net balance of depolarizing and hyperpolarizing ionic currents, with the latter largely mediated by $K^+$ channels (Firth et al., 2011; Makino et al., 2011; Mondéjar-Parreño et al., 2021). Among these, strongly inwardly rectifying $K^+$ (Kir2) channels and the ATP-sensitive $K^+$ ($K_{ATP}$) channels play key roles in $V_M$ regulation across vascular cell types (Nelson & Quayle, 1995; Quayle et al., 1997; Sancho et al., 2022; Sonkusare et al., 2016). In the vascular wall, Kir2 channels are tetrameric assemblies primarily composed of Kir2.1 and Kir2.2 subunits. This molecular architecture confers a distinct biophysical profile marked by strong inward rectification, potentiation by elevated extracellular $K^+$, and robust voltage-dependent block by micromolar concentrations of $Ba^{2+}$ (Sancho et al., 2019, 2024; Sonkusare et al., 2016). In contrast, vascular $K_{ATP}$ channels are octameric complexes formed by four pore-forming Kir6.1 subunits and four regulatory sulfonylurea receptors (SUR2B), with gating modulated by intracellular ATP/ADP levels (Li et al., 2013; Quayle et al., 1994; Sancho et al., 2022). These channels are activated by synthetic openers such as pinacidil and by endogenous vasodilators including adenosine and calcitonin gene-related peptide, and are inhibited by sulfonylureas such as glibenclamide or the putative Kir6.1/SUR2B-selective blocker PNU-37783A (Nelson & Quayle, 1995; Quayle et al., 1994; Sancho et al., 2022).

Despite their recognized physiological relevance in multiple vascular territories – particularly within the cerebral microcirculation (Longden et al., 2017; Sancho et al., 2019, 2022, 2024) – direct functional evidence of Kir2 and $K_{ATP}$ channel activity in native pulmonary artery (PA) SMCs and PA endothelial cells (ECs) remains limited. This gap in knowledge is particularly significant given that alterations in the expression and function of specific $K^+$ channel subfamilies – including voltage-gated (Kv) and two-pore domain $K^+$ (TASK1) channels (Antigny et al., 2016; Mondéjar-Parreño et al., 2021; Saint-Martin Willer et al., 2023; Yuan, Aldinger et al., 1998; Yuan, Wang et al., 1998) – contribute to the onset and progression of pulmonary artery hypertension (PAH), a condition characterized by persistent vaso-constriction and elevated pulmonary vascular resistance (Giaid, 1998; Oka et al., 2007). Moreover, pathogenic variants in *ABCC8*, *ABCC9* and *KCNJ8* genes which encode the SUR1, SUR2 and Kir6.1 subunits of $K_{ATP}$ channels, respectively, have been associated with heritable forms of PAH (Le Ribeuz et al., 2023; Montani et al., 2023). Elucidating the functional roles of Kir2 and $K_{ATP}$ channels in the pulmonary circulation could therefore provide critical insights into the complex mechanistic networks that regulate vascular tone and potentially uncover novel therapeutic targets for PAH and related pulmonary vascular disorders.

In this study, we investigated whether Kir2 and $K_{ATP}$ channels are functionally expressed in human PASMCs and PAECs, and assessed their potential contributions to resting $V_M$ and vascular tone regulation. Experiments were conducted on freshly enzymatically dissociated PASMCs and PAECs, and intact pulmonary resistance arteries, using patch-clamp electrophysiology, immuno-labelling and vessel myography. Electrophysiological recordings confirmed the presence of Kir2 and $K_{ATP}$ currents in both native PASMCs and PAECs. Immuno-histochemical analyses further demonstrated the expression of Kir2.1, Kir2.2, Kir6.1 and SUR2 sub-units in both vascular layers. Functional studies in intact vessels revealed that Kir2 channels exert a tonic influence on pulmonary arterial tone, while $K_{ATP}$ channel activation promotes vasodilatation – underscoring a key electro-metabolic role in the regulation of $V_M$ and vasomotor responses. The study provides the first direct functional evidence of Kir2 and $K_{ATP}$ channel activity in freshly isolated human pulmonary arterial cells and highlights their potential relevance in the control of pulmonary vascular tone and haemodynamics.

## Material and methods

### Human procedures

Non-tumoral lung tissue was collected from 29 patients (16 men and 13 women; mean age 67.1 (8.0) years) under-going lobectomy as part of the surgical treatment for lung carcinoma at the Hospital Universitario de Getafe (Madrid, Spain). Due to the limited number of samples, sex differences were not assessed in the present study. The protocol was approved by the Human Research Ethics Committee of the Hospital Universitario de Getafe (Ref. A04/16) and written informed consent was obtained from all patients for the use of lung tissue discarded by pathologists following thoracic surgery. All procedures were carried out in accordance with the guidelines of the *Declaration of Helsinki*. Clinical records were reviewed to collect information on medication use and smoking history. Some patients were taking cardiovascular or respiratory medications (e.g. anti-hypertensives, statins, inhaled therapies), and a subset has a history of smoking (10 ex-smokers and two current smokers). These factors are inherent to studies using human surgical specimens and may contribute to variability in vascular and electro-physiology responses. Upon arrival at the laboratory, human samples were placed in cold Krebs solution (pH 7.4) containing: 119 mM NaCl, 4.7 mM KCl, 1.2 mM $KH_2PO_4$, 1.2 mM $MgSO_4$, 2 mM $CaCl_2$, 11 mM glucose and 11 mM sodium $NaHCO_3$.

## Isolation of native pulmonary artery smooth muscle and endothelial cells

Single SMCs and ECs were freshly isolated from intact human resistance PAs. Peripheral lung parenchyma was carefully dissected under a stereomicroscope, and arteries were cut longitudinally, and placed in $Ca^{2+}$-free physiological saline solution ($Ca^{2+}$-free PPS) containing: 130 mM NaCl, 5 mM KCl, 1.2 mM MgCl$_2$, 10 mM glucose and 10 mM HEPES (pH 7.4, adjusted with NaOH). Vessels displaying typical arterial morphology – thicker wall, round cross-sectional profile, and well-defined smooth muscle layers – were selected for isolation, while veins (thinner walls, irregular shape, no clear smooth muscle layers) and bronchial structures (irregular wall structure with cartilage and discontinuous smooth muscle layers) were excluded.

PASMCs were enzymatically isolated as previously described (Cogolludo et al., 2009). Briefly, resistance pulmonary arteries were longitudinally opened and incubated for 30 min at 37°C in $Ca^{2+}$-free PPS containing 1.5 mg/ml papain, 0.8 mg/ml dithiothreitol and 0.7 mg/ml albumin. Following enzymatic digestion, tissues were thoroughly washed multiple times with ice-cold $Ca^{2+}$-free PPS and actively triturated using a fire-polished Pasteur pipette to release individual cells. The resulting cells were stored in ice-cold $Ca^{2+}$-free PPS and used within approximately 5 h of isolation. Cells were identified based on their characteristic spindle-shaped morphology and contractile behaviour.

PAECs were isolated using a protocol adapted from Longden et al. (2017) with minor adjustments to enhance cell yield. Briefly, arterial segments (∼2 mm in length) were incubated in $Ca^{2+}$-free PPS containing 0.5 mg/ml neutral protease, 0.5 mg/ml elastase and 100 μM CaCl$_2$ for 60 min at 37°C. Following this incubation, 0.5 mg/ml collagenase type I was added for an additional 5 min at 37°C. Tissues were then washed several times with ice-cold $Ca^{2+}$-free PPS and gently triturated (∼10 times) using a fire-polished glass pipette. This isolation procedure resulted in small EC clusters (endothelial sheets) and individual ECs, which were identified by their rough shape and the absence of voltage-dependent K$^+$ conductances. Isolated PAECs were maintained in ice-cold $Ca^{2+}$-free PPS and used up to a maximum of 4 h.

### Electrophysiological recordings

Conventional patch-clamp electrophysiology was used to measure whole-cell currents in freshly isolated PASMCs and PAECs. Currents were amplified with an Axopatch 200B amplifier (Axon Instruments, Union City, CA), filtered at 1 kHz, digited at 5 kHz (Digidata 1322 A) and stored on a computer for offline analysis using Clampfit 10.7 software (Molecular Devices, San José,

CA). Patch pipettes were pulled from borosilicate micro-capillary tubes (1 mm outer diameter and 0.6 mm inner diameter; Narishige, Japan) and fire-polished to achieve a tip resistance of 4–5 MΩ. All experiments were conducted at room temperature. The cell capacitance for PASMCs ranged from 13 to 21 pF ($n = 20$), and for PAECs, it ranged from 8 to 14 pF ($n = 20$), with values recorded using the cancellation circuit in the voltage-clamp amplifier. Cells exhibiting a significant change in whole-cell capacitance ($>0.3$ pF) during the experiment were excluded from the analysis. The whole-cell configuration was established by gently lowering a micropipette onto a cell, applying negative pressure to form a high-resistance (GΩ) seal, and then rupturing the membrane to allow access to the cell interior.

To record Kir2 currents, cells were voltage-clamped at a holding potential of −50 mV and subjected to a voltage ramp from −140 to +40 mV, over a duration of 400 ms, in the absence and presence of BaCl$_2$ (100 μM), a selective blocker of Kir2 channels within the micromolar range. The bath solution contained 80 mM NaCl, 60 mM KCl, 1 mM MgCl$_2$, 10 mM HEPES, 10 mM glucose and 2 mM CaCl$_2$ (pH 7.4). The pipette solution consisted of 10 mM NaOH, 11.4 mM KOH, 128.6 mM KCl, 1.1 mM MgCl$_2$, 3.2 mM CaCl$_2$, 5 mM EGTA and 10 mM HEPES (300 nM free $Ca^{2+}$; pH 7.2). For K$_{ATP}$ current recordings, cells were exposed to an extracellular solution identical to the one used for Kir2 measurements. Pipettes were backfilled with a solution containing 102 mM KCl, 38 mM KOH, 10 mM NaCl, 1 mM MgCl$_2$, 1 mM CaCl$_2$, 10 mM EGTA, 10 mM glucose, 10 mM HEPES, 0.1 mM ATP and 0.1 mM ADP (pH 7.2). Cells were held at a potential of −70 mV, and to enhance the amplitude of K$^+$ currents at hyper-polarized potentials, the external K$^+$ concentration was increased to 60 mM. In this setup, the K$^+$ current is inward, resulting in downward deflections. K$_{ATP}$ currents were activated by the synthetic compound pinacidil (10 μM) and blocked with the selective blockers glibenclamide (10 μM) or PNU-37883A (10 μM). Membrane potential recordings were performed in current-clamp mode using a bath solution containing 135 mM NaCl, 5 mM KCl, 1 mM MgCl$_2$, 10 mM HEPES, 10 mM glucose and 2 mM CaCl$_2$ (pH 7.4), with the same pipette solution as used for voltage-clamp recordings.

### Immunohistochemistry and immunocytochemistry

Protein expression of Kir2.1, Kir2.2, Kir6.1 and SUR2 was evaluated in isolated PASMCs and in intact endothelial layers obtained from whole-mount pulmonary artery preparations. Cell type identification was performed by labelling with CD31 for ECs and $\alpha$-smooth muscle actin for PASMCs (see Table 1). Isolated PASMCs were air-dried onto polylysine-coated microscope coverslips, whereas

**Table 1. Specifications of primary and secondary antibodies employed**

| Antibody (Ab) | Immunogen | Working dilution | Type | Supplier Cat # | Host/Isotype | [a]RRID |
|---|---|---|---|---|---|---|
| Anti-Kir2.1 (*KCNJ2*) | Peptide (C)NGVPESTSTDTPPDIDLHN, corresponding to amino acid (aa) residues 392–410 of human $K_{ir}2.1$ (Accession P48049). Intracellular, C-terminal region. | 1:100 | Polyclonal | Alomone Labs #APC-026 | Rabbit/ IgG | AB_2040107 |
| Anti-Kir2.2 (*KCNJ12*) | Peptide (C)EVATDRDGRSPQPEHDFDR, corresponding to aa residues 391–409 of rat $K_{ir}2.2$ (Accession P52188). Intracellular, C-terminal region. | 1:100 | Polyclonal | Alomone Labs #APC-042 | Rabbit/ IgG | AB_2040109 |
| Anti-Kir6.1 (*KCNJ8*) | Peptide (C)KRNSMRRNNSMRRSN, corresponding to aa residues 382–396 of rat $K_{ir}6.1$ (Accession Q63664). Intracellular, C-terminal region. | 1:100 | Polyclonal | Alomone Labs #APC-105 | Rabbit/ IgG | AB_2039945 |
| Anti-SUR2 (*ABCC9*) | A synthesized peptide derived from human *ABCC9*, corresponding to aa residues L1269-V1319. (Accession O60706). Intracellular, C-terminal region. | 1:100 | Polyclonal | Thermo Fisher Scientific #PA5-103640 | Rabbit/ IgG | AB_2852974 |
| Anti-Actin, $\alpha$-smooth muscle-Cy3 | N-terminal synthetic decapeptide of $\alpha$-smooth muscle actin (ACTA2). | 1:200 | Monoclonal/Clone 1A4 | Sigma-Aldrich #C6198 | Mouse/ IgG2a | AB_476856 |
| Anti-CD31 (PECAM-1) | Activated microglial cells derived from Lewis rats. | 1:100 | Monoclonal/Clone TLD-3412 | Sigma-Aldrich #MAB1393-I | Mouse/IgG1$\kappa$ | Not assigned |
| Anti-rabbit, Alexa Fluor 568 | Gamma immunoglobins heavy and light chains. | 1:500 | Polyclonal | Thermo Fisher Scientific #A-11036 | Goat/ IgG | AB_10563566 |
| Anti-mouse, Alexa Fluor 488 | Gamma immunoglobins heavy and light chains. | 1:500 | Polyclonal | Thermo Fisher Scientific #A-11001 | Goat/IgG | AB_2534069 |

[a]RRID: research resource identifier.

longitudinally opened arteries were pinned flat onto Sylgard blocks. All samples were rinsed in PBS (137 mM NaCl, 2.7 mM KCl, 10 mM $Na_2HPO_4$, 7.4 mM $KH_2PO_4$, pH 7.4), and plasma membranes were labelled with wheat germ agglutinin (WGA) conjugated to Alexa Fluor 488 (Thermo Fisher Scientific) at 1:100 for 20 min in PASMCs, and at 1:50 for 1 h in *en face* arterial preparations. Samples were then fixed with 4% paraformaldehyde/PBS for 1 h at room temperature (RT), followed by permeabilization and blocking (1 h, RT) using a quench solution consisting of PBS supplemented with 3% goat serum and 0.2% Tween 20. Samples were subsequently incubated overnight at 4°C in a humidified chamber with primary antibodies against Kir2.1, Kir2.2, Kir6.1 and SUR2 (Table 1), diluted in antibody buffer (PBS containing 2% goat serum and 0.2% Tween 20). The next day, samples were washed three times (5 min each) with PBS-0.2% Tween 20 and incubated with either Alexa Fluor 568-conjugated goat anti-rabbit IgG (H+L) or Alexa Fluor 488-conjugated goat anti-mouse IgG (H+L) secondary antibody (1:500)

for 2 h at RT, protected from light. After additional washes, nuclei were stained with DAPI (1 µg/ml in PBS) for 20 min at RT in the dark. The preparations were then washed again and mounted onto glass slides using an antifade mounting medium (ProLong Gold, Thermo Fisher), with whole-mount preparations oriented so that the endothelial layer faced upward. Control samples, processed in parallel, were incubated without primary antibodies. Fluorescence images were captured using a ×60 oil immersion lens on an Olympus FV1200 spectral confocal laser-scanning microscope, equipped with the appropriate filter sets. The images were saved as 12-bit digital files using Olympus FluoView-1200 software. Z-stack images (*z*-step 1 µm) were acquired and imported into ImageJ for further analysis, where each stack was collapsed into a maximal-intensity projection. Imaging was performed at the Microscopy and Cytometry Centre (CAI), Complutense University of Madrid, Spain.

### Pulmonary artery reactivity

Immediately upon collection, human lung tissue was immersed in ice-cold (4°C) Krebs solution, and PAs were subsequently isolated as previously described (Pandolfi et al., 2017). PA rings (2 mm in length, 0.5–0.8 mm internal diameter) were mounted in wire myograph chambers (610M, Danish Myo Technology, Aarhus, Denmark) filled with Krebs buffer solution maintained at 37°C and continuously bubbled with a gas mixture of 21% $O_2$, 5% $CO_2$ and 74% $N_2$. For each artery, the passive wall tension–internal circumference relationship was determined, and the internal circumference corresponding to a transmural pressure of 30 mmHg for a relaxed vessel *in situ* (L30) was calculated (Cogolludo et al., 2009). Arteries were then normalized to an internal circumference (L1) equal to 0.9 × L30, a dimension at which force development is near maximal (Mulvany & Halpern, 1977). Following normalization, arterial segments were maintained at a transmural pressure of 30 mmHg during the equilibration period prior to functional assessment (Ozaki et al., 1998), reflecting the physiological pressure experienced *in situ*. After stabilization, PA rings were depolarized with KCl (80 mm; equimolar replacement of $Na^+$) to determine their viability and contractile capacity, then thoroughly washed to ensure full recovery before subsequent stimulations. Vessels that did not respond to the KCl challenge (less than 0.25 mN/mm$^2$) were excluded from the analysis. After an additional 20 min equilibration period, contractile responses to cumulative concentrations of $BaCl_2$ (3–100 µM) were assessed in the absence and in the presence of nifedipine (1 µM). PA rings were randomly assigned to either time control or $Ba^{2+}$ treatment groups to avoid bias, and contractile

responses to $BaCl_2$ were expressed as a percentage of the maximal contraction induced by high [$K^+$] (80 mm). Endothelium-dependent vasodilatory responses to acetylcholine (ACh; 0.001–100 µM) were evaluated in PA rings precontracted with the thromboxane $A_2$ analogue U46619 (0.1 µM), in the absence or the presence of $BaCl_2$ (100 µM, added 20 min before U46619 and maintained throughout the ACh dose-response curve). Rings exhibiting <20% relaxation to 100 µM ACh were classified as endothelium-compromised and excluded. In a separate set of experiments, endothelium-independent relaxation to sodium nitroprusside (SNP; 0.01 nm to 10 µM) was assessed before and after $BaCl_2$ (100 µM) to test smooth muscle responsiveness to NO. In addition, cumulative vasodilatory responses to pinacidil (0.03–100 µM) were tested in U46619 (0.1 µM)-precontracted arteries before and after 20 min incubation with either vehicle (DMSO), glibenclamide (10 µM) or PNU-37883A (30 µM). Relaxation responses were expressed as a percentage of the maximal contraction achieved with U46610.

### Chemicals

Unless stated otherwise, all buffers, chemicals, and reagents utilized in the study were obtained from Merck (Darmstadt, Germany). Enzymes including papain, neutral protease, elastase and collagenase I were from Worthington, and PNU-37883A from Tocris (Bristol, UK). Primary antibodies against Kir2.1, Kir2.2 and Kir6.1 were from Alomone Labs; SUR2 antibody and the membrane marker WGA from Thermo Fisher Scientific (Invitrogen); and CD31 and $\alpha$-smooth muscle actin from Sigma-Aldrich. Secondary antibodies, Alexa Fluor 564 goat anti-rabbit IgG and 488 goat anti-mouse IgG were also from Thermo Fisher Scientific (Invitrogen). Stock solutions of pinacidil, glibenclamide, PNU-37883A and nifedipine were prepared in DMSO and subsequently diluted in distilled water. The final DMSO concentration never exceeded 0.1% in the myograph chambers and did not affect smooth muscle tone in control experiments.

### Statistical analysis

Data are expressed as means $\pm$ standard deviation (SD), with *n* representing the number of cells, arteries or patients included in the analysis. No more than two different experiments were performed on vessels from a given patient. Power analysis was performed using G*Power 3.1.9.7 (Universität Mannheim, Germany), based on previously published data and pilot results, assuming 80% power and $\alpha = 0.05$. Owing to the limited availability of human pulmonary tissue, the number of viable arterial rings or isolated cells varied across experiments. However, final sample sizes remained within ranges

commonly reported in comparable studies. In the myography experiments, vasodilatory responses are expressed as the percentage reversal of the previous constriction induced by U46619 in each artery. Brown-Forsythe and Shapiro-Wilk tests were used to assess the equality of group variances and normal distribution of data, respectively. Where appropriate, paired, or unpaired *t* tests, or two-way ANOVA followed by Bonferroni's multiple comparisons test, were performed to compare the effects of a given treatment on whole-cell current or isometric tension. *P* values less than 0.05 were considered statistically significant, as indicated by stars in the figure panels. All analyses were conducted using Clampfit 10.7 software (Molecular Devices, USA) and GraphPad Prism 8 (GraphPad Software).

## Results

### Human pulmonary artery smooth muscle cells express functional Kir2 channels

To assess the presence of functional strongly inwardly rectifying $K^+$ (Kir2) channels in freshly isolated PASMCs from human pulmonary arteries, we employed the conventional whole-cell configuration of the patch-clamp technique – an approach previously shown to reveal Kir2 channel activity in other vascular cells. Cells were voltage-clamped at a holding potential of −50 mV, dialysed with an internal solution containing 140 mM $K^+$, and perfused with a high-$K^+$ (60 mM $K^+$) external solution to enhance Kir2-mediated currents (Fig. 1*A* and *B*). Whole-cell currents were elicited by 400 ms voltage ramps from −140 to +100 mV and recorded before and after application of $BaCl_2$ (100 μM), a selective Kir2 pore blocker effective at micromolar concentrations (Quayle et al., 1993). Under this experimental setup, $Ba^{2+}$ selectively inhibited inward currents at membrane potentials negative to the calculated potassium equilibrium potential ($E_K = −23$ mV), as determined by the Nernst equation. In contrast, outward currents – primarily mediated by voltage-gated $K^+$ channels – remained largely unaffected by $BaCl_2$ (Fig. 1*B* and *C*). The $Ba^{2+}$-sensitive current component displayed the hallmark inward rectification characteristic of Kir2 channels (Fig. 1*D*). On average, $Ba^{2+}$ reduced peak inward current at -140 mV by ∼75%, yielding a mean $Ba^{2+}$-sensitive current of −8.9 (3.0) pA/pF (Fig. 1*E*). To determine the contribution of Kir2 channels to the resting $V_M$, current-clamp recordings were performed using physiological extracellular [$K^+$]. Compared with time-matched controls, superfusion with $BaCl_2$ (100 μM) induced a robust depolarization from −44.9 (4.7) to –30.4 (4.5) mV within ∼20 min (Fig. 1*F* and *G*). Supporting these electrophysiological data, immunocytochemistry confirmed the expression of Kir2.1 (Fig. 1*I*) and Kir2.2 (Fig. 1*J*) sub-

units $\alpha$-smooth muscle actin-positive PASMCs (Fig. 1*H*). Both subunits showed clear colocalization with the plasma membrane marker WGA (see insets in Fig. 1*I* and *J*), indicating membrane localization. No signal was detected in control samples lacking the primary antibodies. Together, these findings demonstrate that human PASMCs express functional Kir2 channels, likely formed by Kir2.1 and Kir2.2 subunits, which contribute to the regulation of resting $V_M$.

### Human pulmonary artery endothelial cells express functional Kir2 channels

The expression of functional Kir2 channels in native human PAECs was evaluated using whole-cell patch-clamp electrophysiology under the same experimental conditions applied to PASMCs. PAECs were held at −50 mV, and membrane currents were monitored in response to a 400 ms ramp from −140 to +40 mV (Fig. 2*A*). Like PASMCs, endothelial cells displayed a $Ba^{2+}$-sensitive, nonlinear inward current that activated at potentials more hyperpolarized than the experimentally determined $E_K$. However, unlike PASMCs, PAECs showed minimal outward currents at membrane potentials positive to 0 mV, suggesting a limited contribution of voltage-gated $K^+$ channels to the overall $K^+$ conductance in these cells (Fig. 2*A* and *B*). $Ba^{2+}$-subtracted currents exhibited the typical electrophysiological signature of Kir2-mediated currents (Fig. 2*C*). Upon hyperpolarization, PAECs developed robust inward currents, with peak current densities reaching −21.4 (8.0) pA/pF at −140 mV. $Ba^{2+}$ reduced this current by ∼83% resulting in a $Ba^{2+}$-sensitive current with a mean amplitude of −17.8 (8.2) pA/pF (Fig. 2*D*). Notably, Kir2 currents in PAECs were approximately twofold larger than those recorded in PASMCs under identical experimental conditions. For visualization of endothelial channel expression, *en face* preparations were imaged by confocal microscopy with *z*-stack acquisition. Under these conditions, the endothelial layer was clearly identified by CD31 immunoreactivity and characteristic morphology – including rounder nuclei with a slightly irregular distribution – followed by the internal elastic lamina and underlying PASMC layers, distinguished by elongated nuclei oriented perpendicularly to those of the endothelium (Fig. 2*E*). Immunofluorescence analysis revealed robust Kir2.1 and Kir2.2 labelling in the endothelial compartment (Fig. 2*F* and *G*). Notably, Kir2 signals colocalized with WGA staining (insets in Fig. 2*F* and *G*), supporting their plasma membrane localization. No specific signal was detectable in control samples lacking the primary antibody. Collectively, these findings indicate that human pulmonary artery PAECs express functional Kir2 channels.

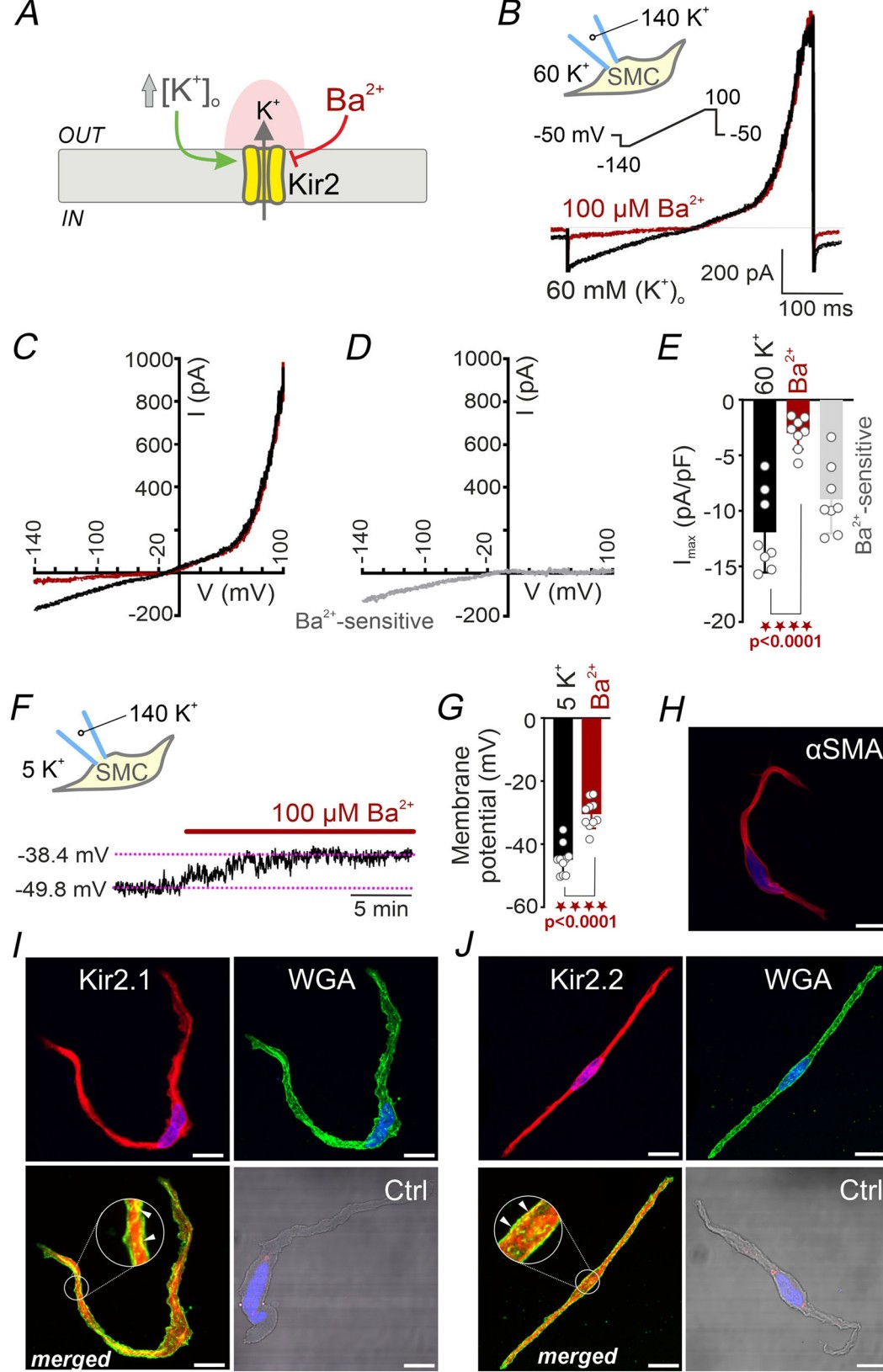

**Figure 1. Human pulmonary artery smooth muscle cells (PASMCs) express functional Kir2 channels**

*A*, schematic representation of Kir2 channel activation by increased K$^+$ and inhibition by Ba$^{2+}$. *B*, representative whole-cell recording from a freshly isolated PASMC bathed in high-K$^+$ external solution (60 mM) and dialysed

with an internal solution containing 140 mM $K^+$ (inset), before (black) and after (dark red) application of 100 µM $Ba^{2+}$. Cells were subjected to a voltage ramp protocol (−140 to +100 mV from a holding potential of -50 mV). *C*, current–voltage (*I–V*) relationships obtained under control and $Ba^{2+}$ conditions from the same cell. *D*, $Ba^{2+}$-sensitive current, exhibiting the characteristic inward rectification of Kir2-mediated currents. *E*, summary data comparing peak (−140 mV) inward current density (pA/pF) of control (60 $K^+$), $Ba^{2+}$ and $Ba^{2+}$-sensitive currents (*n* = 8 cells from five patients). *F*, representative recording of resting $V_M$ from a PASMC bathed in physiological extracellular $[K^+]$ (5 mM), before and after superfusion with $BaCl_2$ (100 µM). *G*, summary of resting $V_M$ values in the absence and presence of $Ba^{2+}$ (*n* = 8 cells from three patients). Data are presented as means (SD); *P* values indicate comparison *vs*. control paired Student's *t* test. *H*, PASMC identity confirmed by positive *α*-smooth muscle actin immunolabeling. *I*, immunocytochemical detection of Kir2.1 (red, *top left*), and WGA (plasma membrane marker; green, *top right*), with merged image (*bottom left*) and corresponding negative control (Ctrl; *bottom right*) in native PASMCs. Arrowheads indicate examples of marker colocalization. *J*, equivalent immunostaining for Kir2.2. Nuclei were counterstained with DAPI (blue) in all images. Each experiment was performed using cells from four independent patients; images are representative of 15–20 cells per patient. Scale bars = 10 µm.

## Kir2 channels are key regulators of human pulmonary vascular tone

To elucidate the contribution of Kir2 channels to the regulation of human PA tone, isometric tension measurements were conducted in isolated PA rings mounted on wire myographs (Fig. 3*A*). Cumulative addition of $BaCl_2$ (3–100 µM), within a concentration range known to selectively inhibit Kir2 channels, induced a robust, concentration-dependent contractile response (Fig. 3*B*). Compared with time-matched controls, $BaCl_2$ significantly increased vascular tone, reaching 14.1 (9.6)% and 35.0 (21.8)% of the maximal KCl-induced contraction at 30 µM and 100 µM, respectively (Fig. 3*C*). In a separate set of experiments, $BaCl_2$-induced contractions were blunted in the presence of nifedipine (1 µM), supporting the notion that Kir2 channel inhibition leads to membrane depolarization, which in turn enhances $Ca^{2+}$ influx via L-type $Ca^{2+}$ channels. These findings demonstrate that Kir2 channels exert a tonic hyperpolarizing influence that limits basal pulmonary vasoconstriction, and that $BaCl_2$-induced contractions are driven by a depolarization-dependent mechanism mediated through L-type $Ca^{2+}$ channel activation.

To further explore the functional relevance of Kir2 channels in the endothelium, endothelium-dependent relaxation responses to cumulative doses of ACh (0.001–100 µM) were evaluated in PA rings precontracted with U46619 (0.1 µM), in the absence and presence of $BaCl_2$ (100 µM). Under control conditions, ACh induced a concentration-dependent relaxation (Fig. 3*D*), reaching a maximum of 42.9 (22.91)% at 100 µM (Fig. 3*F*), consistent with previous studies using this preparation (Pandolfi et al., 2017). $BaCl_2$ significantly attenuated this response (Fig. 3*E*), with maximal relaxation reduced to 19.6 (23.2)% (Fig. 3*F*). To rule out effects on smooth muscle sensitivity to nitric oxide (NO), additional experiments evaluated SNP-induced relaxation (0.1 nM to 10 µM) in the absence and presence of $BaCl_2$ (100 µM). SNP showed a similar vasodilatory efficacy as ACh, reaching a maximum of 36.0 (14.6)% and 37.3 (17.2)% at 10 µM under control and $BaCl_2$-treated conditions, respectively (Fig. 3*G*

and *H*). No significant differences in SNP-induced vasodilatation were observed between both groups, indicating preserved NO responsiveness. Collectively, these findings demonstrate that Kir2 channels contribute to the maintenance of basal pulmonary vascular tone and play a critical role in mediating endothelium-dependent vasodilatation in human pulmonary arteries.

## Functional Kir6.1/SUR2B $K_{ATP}$ channels are expressed in human pulmonary artery smooth muscle cells

We next investigated whether native PASMCs from human PAs exhibit functional $K_{ATP}$ channel activity. To this end, conventional whole-cell patch-clamp recordings were performed under conditions optimized to isolate $K_{ATP}$-mediated currents. Cells were held at a membrane potential of −70 mV, exposed to a high-$K^+$ (60 mM) external solution, and dialysed with an internal solution containing 140 mM $K^+$, low ATP (0.1 mM) and a relatively high ADP (0.1 mM). This ionic and metabolic setup maximized the driving force for $K^+$ ions, promoting inward $K^+$ influx (Quayle et al., 1994) while minimizing interference from other channel types. Under these conditions, application of the $K_{ATP}$ channel opener pinacidil (10 µM) evoked marked inward currents (Fig. 4*A*–*C*). This effect was consistently suppressed upon the subsequent addition of either the sulfonylurea glibenclamide (10 µM), a classic $K_{ATP}$ channel blocker (Fig. 4*B*), or PNU-37883A (10 µM), a more selective inhibitor of vascular-type Kir6.1/SUR2B $K_{ATP}$ channels (Fig. 4*C*). Quantitative analysis revealed that pinacidil increased current density to −15.3 (9.1) pA/pF, while glibenclamide reduced this current to −1.4 (1.7) pA/pF, yielding a glibenclamide-sensitive component of −13.9 (8.1) pA/pF (Fig. 4*B* and *D*). Similarly, PNU-37883A reduced pinacidil-evoked currents to −1.0 (0.8) pA/pF, with a PNU-sensitive component of −11.2 (4.5) pA/pF (Fig. 4*C* and *E*), confirming the functional involvement of Kir6.1/SUR2B-containing $K_{ATP}$ channels in these cells. Immunocytochemistry further confirmed the molecular identity and membrane localization of these channels,

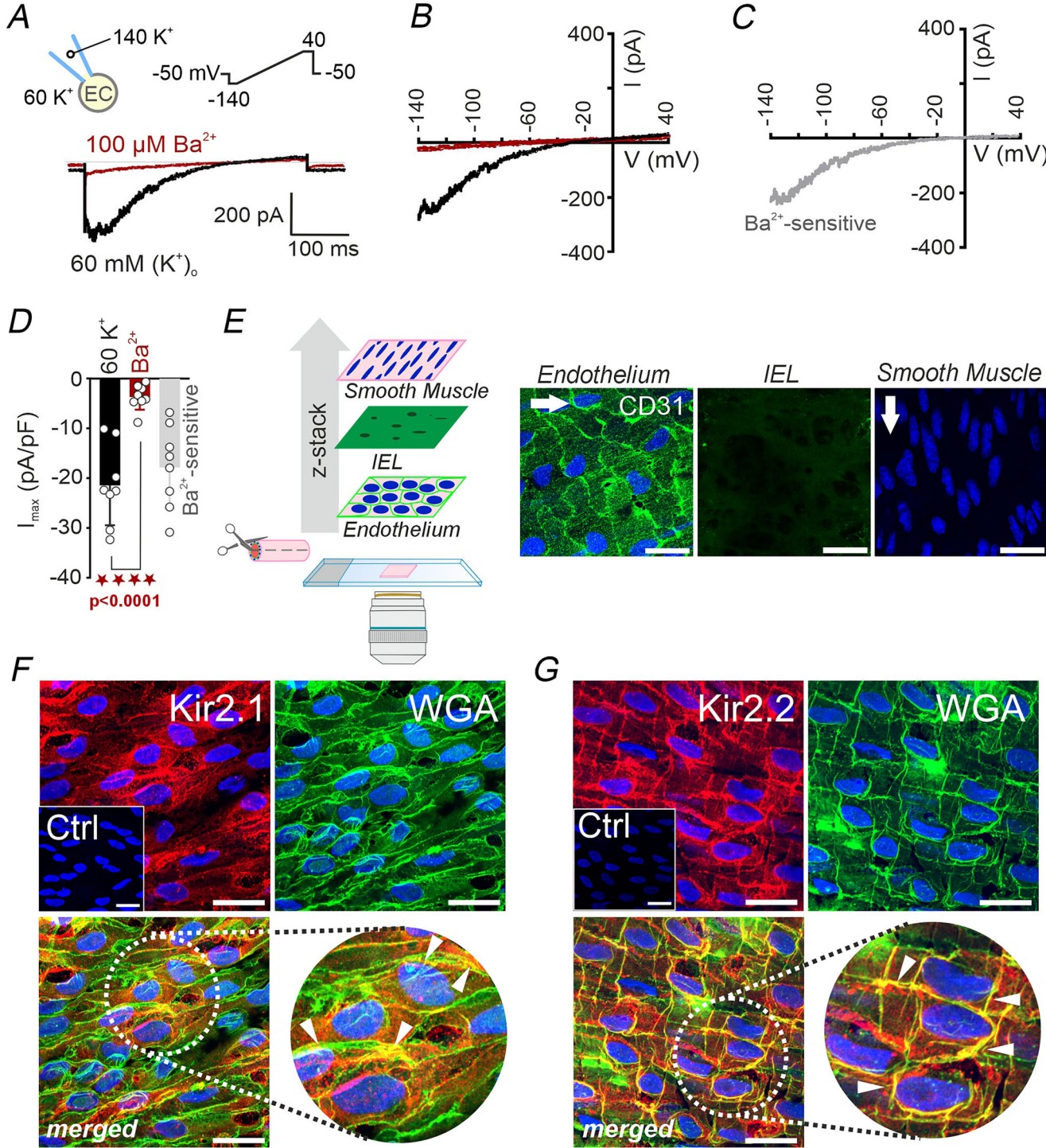

**Figure 2. Human pulmonary artery endothelial cells (PAECs) express functional Kir2 channels**

*A*, representative whole-cell current traces recorded from a freshly isolated PAEC bathed in high extracellular K$^+$ (60 mM) and dialysed with an internal solution containing 140 mM K$^+$, before (black) and after (dark red) application of 100 μM Ba$^{2+}$ (inset). Cells were subjected to a voltage ramp protocol (−140 to +100 mV from a holding potential of −50mV). *B*, corresponding current–voltage (*I–V*) relationships under control and Ba$^{2+}$ conditions. *C*, Ba$^{2+}$-sensitive current, displaying the hallmark inward rectification characteristic of Kir2 channels. *D*, summary data of peak inward current density (pA/pF) at −140 mV under control conditions (60 K$^+$), following Ba$^{2+}$ application, and for the Ba$^{2+}$-sensitive component. Data are presented as means (SD), paired Student's *t* test (*n* = 8 cells from five patients). *E*, schematic of the *en face* preparation procedure and representative confocal

*z*-stack images showing endothelial cell identification. The endothelial layer was recognized by CD31 immuno-reactivity and round nuclei with slightly irregular distribution, followed by the internal elastic lamina (IEL) and underlying PASMC layers characterized by elongated nuclei oriented perpendicularly to those of the endothelium. *F*, *en face* immunofluorescence staining of Kir2.1 (red, *top left*) and the plasma membrane marker WGA (green, *top right*); merged image (*bottom left*) with a magnified inset (*bottom right*). Corresponding negative control (Ctrl) is shown as inset. Arrowheads indicate examples of marker colocalization. *G*, equivalent immunostaining for Kir2.2. Nuclei were counterstained with DAPI (blue) in all images. Each experiment was performed using cells from four independent patients; images are representative of 15–20 cells per patient. Scale bars = 20 μm.

revealing strong staining of the pore-forming Kir6.1 subunit (Fig. 4*F*) and the regulatory subunit SUR2 (Fig. 4*G*) in PASMCs, both of which colocalized with the membrane marker WGA (insets in Fig. 4*F* and *G*). Omission of the primary antibody abolished the signal, confirming the specificity of the staining. These findings demonstrate that human PASMCs express functional $K_{ATP}$ channels comprising Kir6.1 and SUR2 subunits.

## Human pulmonary artery endothelial cells express functional Kir6.1/SUR2B $K_{ATP}$ channels

The expression of functional $K_{ATP}$ channels in native human PAECs was examined using whole-cell patch-clamp recordings, with experimental conditions matching those applied to PASMCs. PAECs were held at −70 mV and dialysed with an internal solution containing a low ATP to ADP ratio (0.1 mM each) to promote channel activation. Under these conditions, the application of pinacidil (10 μM) triggered robust inward currents (Fig. 5*A* and *B*). This effect was effectively reverted by subsequent addition of either glibenclamide (10 μM, Fig. 5*A*) or PNU-37883A (10 μM, Fig. 5*B*). Quantitative analysis revealed that pinacidil augmented current density to −22.9 (4.6) pA/pF, while glibenclamide reduced the current to −4.2 (1.4) pA/pF, corresponding to a glibenclamide-sensitive current component of −18.7 (3.9) pA/pF (Fig. 5*C*). Likewise, PNU-37883A diminished the pinacidil-evoked current to −2.6 (1.6) pA/pF, yielding a PNU-sensitive component of -15.5 (2.9) pA/pF (Fig. 5*D*). Immunofluorecence analysis of intact endothelial layers in *en face* preparations further confirmed the expression of Kir6.1 and SUR2 subunits (Fig. 5*E* and *F*), with signals colocalizing with the membrane marker WGA. No specific staining was observed in parallel control experiments where the respective primary antibody was omitted. Together, these data provide compelling evidence that native human PAECs express functional, pharmacologically responsive Kir6.1/SUR2B $K_{ATP}$ channels.

## Activation of $K_{ATP}$ channels induces vasodilatation in human pulmonary arteries

The potential contribution of $K_{ATP}$ channels to human PA tone regulation was evaluated using wire myography on isolated PA rings. Vessels were precontracted with U46619 (0.1 μM), followed by cumulative administration of the $K_{ATP}$ channel opener pinacidil (0.03–30 μM). Under vehicle-treated conditions, pinacidil induced a concentration-dependent relaxation (Fig. 6*A*), with a maximal response of 53.2 (26.1)% (30 μM; Fig 6*D*). This effect was significantly attenuated by pretreatment with either glibenclamide (10 μM; Fig. 6*B*) or PNU-37883A (30 μM; Fig. 6*C*), reducing the maximal relaxation to 24.9 (14.3)% and 29.7 (21.4)%, respectively (Fig 6*D*). Both inhibitors caused a marked rightward shift in the concentration-response curve and a reduction in the maximal relaxation at the concentrations tested (Fig. 6*D*). Importantly, neither $K_{ATP}$ channel blocker affected U46619-induced contractions. These results indicate that functional $K_{ATP}$ channels play a significant role in mediating vasodilatation in human PAs and may represent a pharmacological target for modulating pulmonary vascular tone.

## Discussion

The structural organization of pulmonary resistance arteries – comprising an inner endothelial monolayer ensheathed by concentric layers of contractile PASMCs – enables precise control of vascular tone in response to a dynamic range of stimuli (Segal & Duling, 1986). These two cell types are electrically and functionally coupled via myoendothelial gap junctions, enabling bidirectional communication critical for matching regional blood flow to tissue demands (Mironova et al., 2024; Welsh et al., 2018). A key determinant of this functional inter-face is the resting $V_M$, which integrates the activity of multiple ionic conductance to modulate vascular tone (Nelson et al., 1990). In the pulmonary circulation, this bioelectrical attribute maintains the characteristically low vascular resistance under normoxic conditions and mediates adaptive responses to both physiological and pathological stimuli (Evans et al., 1998; Sylvester et al., 2012).

In both PASMCs and PAECs, $V_M$ is determined by the interplay of inward and outward ionic currents, primarily driven by $K^+$ channels (Firth et al., 2011; Makino et al., 2011; Mondéjar-Parreño et al., 2021). Activation of $K^+$ channels leads to $K^+$ efflux and membrane hyper-polarization, reducing the open probability of L-type

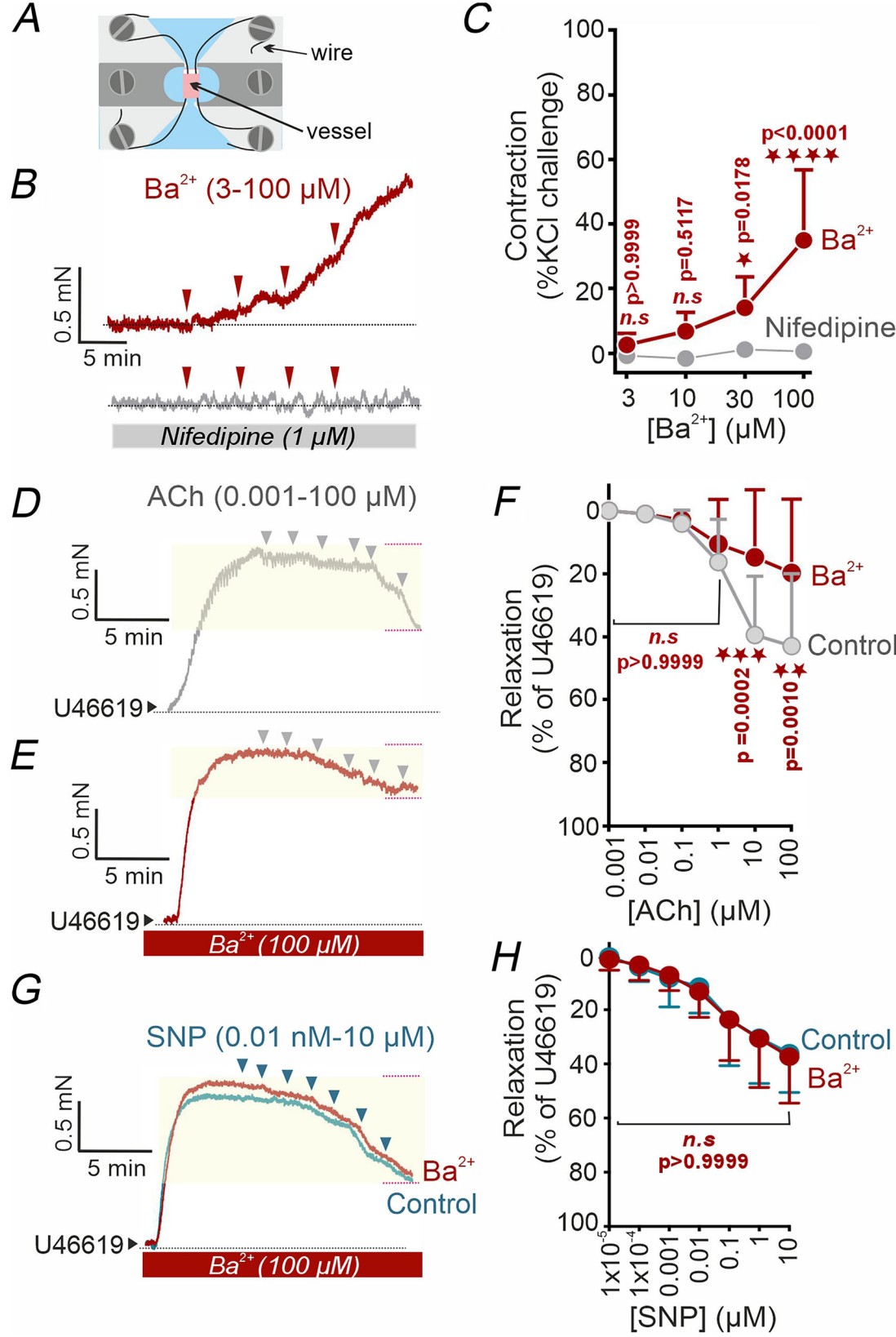

**Figure 3. Kir2 channels modulate basal tone and endothelium-dependent vasodilatation in human pulmonary arteries (PAs)**

*A*, schematic of the wire myograph setup used for isometric tension recordings in isolated PA rings. *B*, representative traces showing BaCl$_2$-induced contraction following cumulative application of Ba$^{2+}$ (3–100 μM) in the absence

(*top*) or presence of nifedipine (1 µM; *bottom*). *C*, concentration-response curve for $BaCl_2$-induced contraction, expressed as % of maximal KCl-induced contraction, showing a dose-dependent increase in vascular tone ($n = 15$ PA rings from five patients), which was attenuated by nifedipine pretreatment. *D* and *E*, representative traces of endothelium-dependent relaxation responses to cumulative doses of ACh (0.001–100 µM) in PA rings pre-contracted with U46619 (0.1 µM), under control conditions (*D*) or in the presence of $BaCl_2$ (100 µM) (*E*). *F*, summary data showing that $BaCl_2$ significantly blunted ACh-induced relaxation ($n = 9$–10 PA rings from four patients). *G*, representative traces of relaxation responses induced by cumulative doses of SNP (0.01 nM to 10 µM) in PA rings precontracted with U46619 (0.1 µM) under control conditions (blue trace) or in the presence of $BaCl_2$ (100 µM; dark red trace). *H*, summary data showing no significant effect of $BaCl_2$ on SNP-induced relaxation ($n = 13$ PA rings from six patients). Data are presented as means (SD); exact *P* values indicate comparisons *vs.* control (two-way ANOVA followed by Bonferroni's multiple comparisons test).

voltage-gated $Ca^{2+}$ channels (VGCC), decreasing cytosolic $Ca^{2+}$ levels, and thereby promoting vasodilatation (Nelson & Quayle, 1995). Among the major classes of vascular $K^+$ channels, several subtypes – including Kv1.2, Kv1.5, Kv2.1, TASK1 and large-conductance $Ca^{2+}$-activated $K^+$ ($BK_{Ca}$) channels, have been extensively investigated in the pulmonary circulation, particularly in the context of hypoxic pulmonary vasoconstriction and PAH (Antigny et al., 2016; Cogolludo et al., 2009; Firth et al., 2011; Makino et al., 2011; Mondéjar-Parreño et al., 2021; Yuan, Aldinger et al., 1998; Yuan, Wang et al., 1998). However, a comprehensive understanding of the specific $K^+$ channel subtypes responsible for setting and stabilizing the resting $V_M$ in human pulmonary vascular cells, particularly within the endothelium, remains incomplete.

Kir2 channels, with their characteristic inward rectification at $V_M$ negative to $E_K$ and activation by external $K^+$, are ideally suited to set the resting $V_M$ in vascular cells (Nelson & Quayle, 1995; Nelson et al., 1990; Quayle et al., 1993). Although Kir2-mediated currents have been described in bovine cultured pulmonary SMCs (Tennant et al., 2006) and ECs (Kamouchi et al., 1997; Shimoda et al., 2002), evidence of their functional expression in native human pulmonary vascular cells remained elusive. Importantly, culture conditions can markedly alter ion channel expression and function (Manoury et al., 2009), underscoring the need to assess Kir2 channel activity directly in freshly isolated cells. This is particularly relevant in the human setting, where confirming functional channel expression under native conditions is critical to avoid misleading interpretations derived from culture-induced phenotypic changes. Here, we demonstrate for the first time that freshly isolated human PASMCs and PAECs exhibit robust, $Ba^{2+}$-sensitive inward currents with classical Kir2 biophysical properties. The amplitude and rectification profile of these currents are comparable to those reported in murine brain and mesenteric vessels (Longden et al., 2017; Sancho et al., 2019; Sonkusare et al., 2016). Immunohistochemistry further confirmed prominent expression of Kir2.1 and Kir2.2 subunits in both cell types, implicating these isoforms in the observed $Ba^{2+}$-sensitive conductance and echoing prior findings

in cerebral resistance arteries (Sancho et al., 2019). Interestingly, Kir2 currents were nearly twice as large in PAECs compared with PASMCs, suggesting a potentially dominant role in endothelial function. Moreover, $BaCl_2$ not only induced a concentration-dependent contraction and significantly reduced ACh-evoked, endothelium-dependent relaxation in human PA rings, but also caused depolarization of PASMCs, indicating that Kir2 channels contribute to resting membrane potential maintenance and, consequently, to both basal tone and endothelial vasodilatory signalling.

Our results align with previous work in mesenteric arteries, where endothelial Kir2 channels boost ACh-induced hyperpolarization and vasodilatation through a pathway involving TRPV4-mediated $Ca^{2+}$ influx and inositol trisphosphate ($IP_3$) receptor-mediated $Ca^{2+}$ release from the endoplasmic reticulum. These localized $Ca^{2+}$ signals activate intermediate- and small-conductance $K^+$ (IK and SK, respectively) channels, driving membrane hyperpolarization and vasodilatation (Sonkusare et al., 2012, 2016). Notably, EC-specific deletion of Kir2.1 blunted this response, underscoring its key role in endothelium-dependent signalling. However, the pulmonary circulation may differ in key aspects as in mouse pulmonary arteries, TRPV4 activation did not couple to IK/SK channels, but instead predominantly activated endothelial nitric oxide synthase (eNOS) (Ottolini et al., 2020). These vascular bed-specific differences, along with potential interspecies variability, highlight the need to directly investigate these signalling pathways in human tissues.

Ion channel expression in the vasculature is often compartmentalized to either ECs or SMCs (Jackson, 2005). Our findings challenge this paradigm by revealing robust, functional expression of Kir2 channels in both cell types. This dual distribution suggests that Kir2 may operate in a coordinated manner across the vascular wall to regulate $V_M$ and tone. A precedent for such an arrangement exists in cerebral resistance arteries, where spatially distinct pools of Kir2.1/2.2 channels – one in ECs and another in SMCs – respond selectively to different haemodynamic stimuli, such as shear stress and intravascular pressure, respectively (Sancho et al., 2019). Such spatial organization may also be relevant in

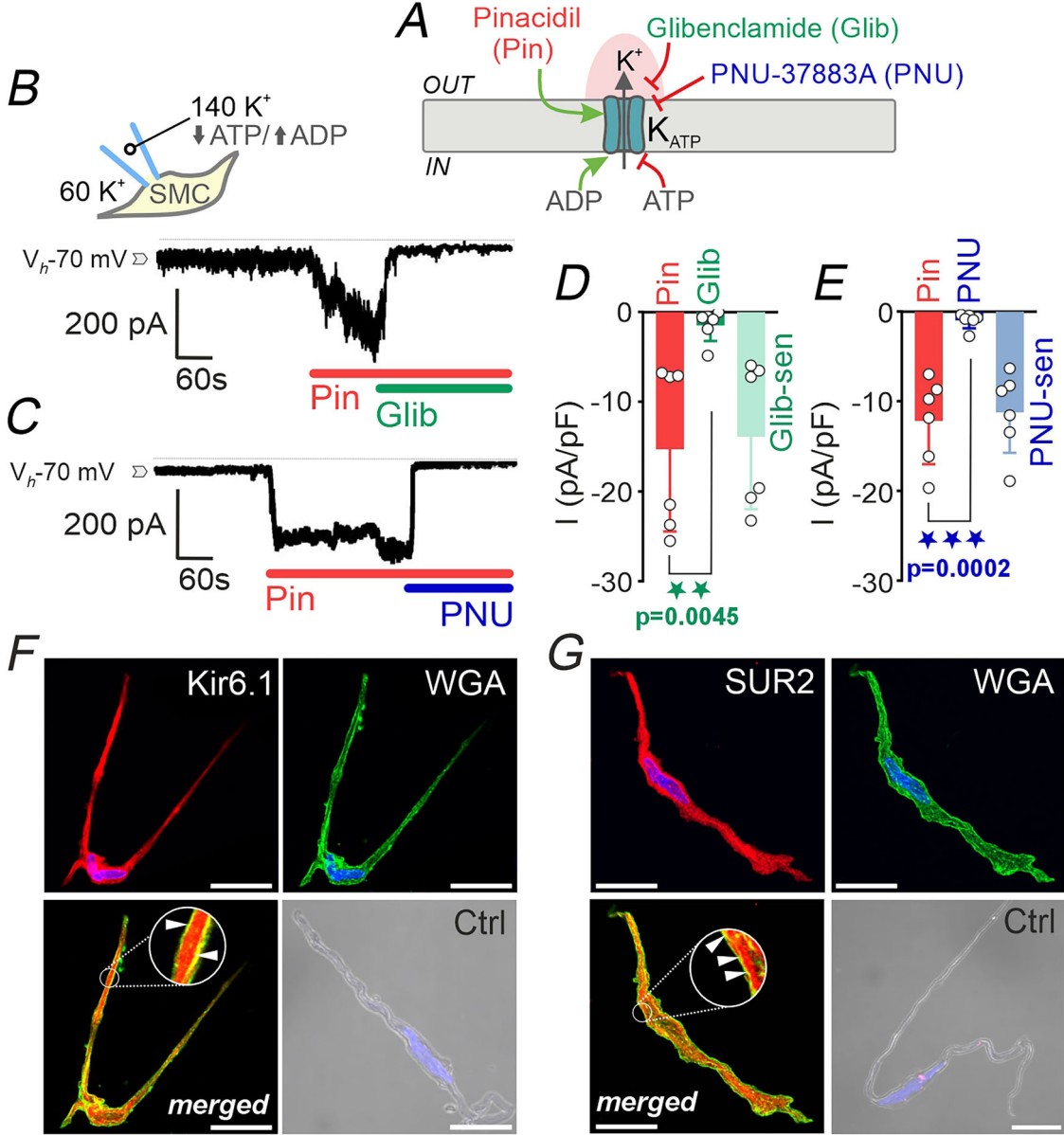

**Figure 4. Functional and molecular expression of K$_{ATP}$ channels in human pulmonary artery smooth muscle cells (PASMCs)**

*A*, schematic diagram of the K$_{ATP}$ channel complex, illustrating its regulation by intracellular nucleotides (ATP/ADP) and pharmacological agents: the channel opener pinacidil (Pin, red), and the blockers glibenclamide (Glib, green) and PNU-37883A (PNU, blue). *B* and *C*, representative time-course recordings of whole-cell currents in freshly isolated PASMCs voltage-clamped at -70 mV in the presence of high extracellular K$^+$ (60 mM) and dialysed with a 140 mM K$^+$ pipette solution with low ATP and elevated ADP levels to facilitate K$_{ATP}$ channel opening. Pinacidil (10 μM) induced robust inward K$^+$ currents, which were subsequently inhibited by glibenclamide (10 μM, *B*) or PNU-37883A (10 μM, *C*). *D* and *E*, summary data showing current density (pA/pF) in response to pinacidil and its inhibition by glibenclamide (*D*; *n* = 6 cells, from five patients) or PNU-37883A (*E*; *n* = 6 cells, from five patients). Data are presented as means (SD), paired Student's *t* test. *F*, immunofluorescence detection of the Kir6.1 subunit (red, *top left*), and the plasma membrane marker WGA (green, *top right*); merged image (bottom left) and negative control (Ctrl; *bottom right*) in native PASMCs. Arrowheads indicate examples of marker colocalization. *G*, equivalent immunodetection for the SUR2 subunit. Nuclei were counterstained with DAPI (blue) in all images. Each experiment was performed using cells from four independent patients; images are representative of 15–20 cells per patient. Scale bars = 10 μm.

the pulmonary circulation, where matching perfusion to ventilation requires constant, finely tuned adjustments in vascular resistance (Glenny & Robertson, 2011). Furthermore, since endothelial Kir2 channels mediate flow-dependent, NO-driven vasodilatation in human mesenteric arteries (Ahn et al., 2017), it is plausible that analogous mechanisms are engaged in the pulmonary endothelium. These mechanisms could have important implications for conditions such as PAH, where impaired vascular tone and endothelial dysfunction are prominent features.

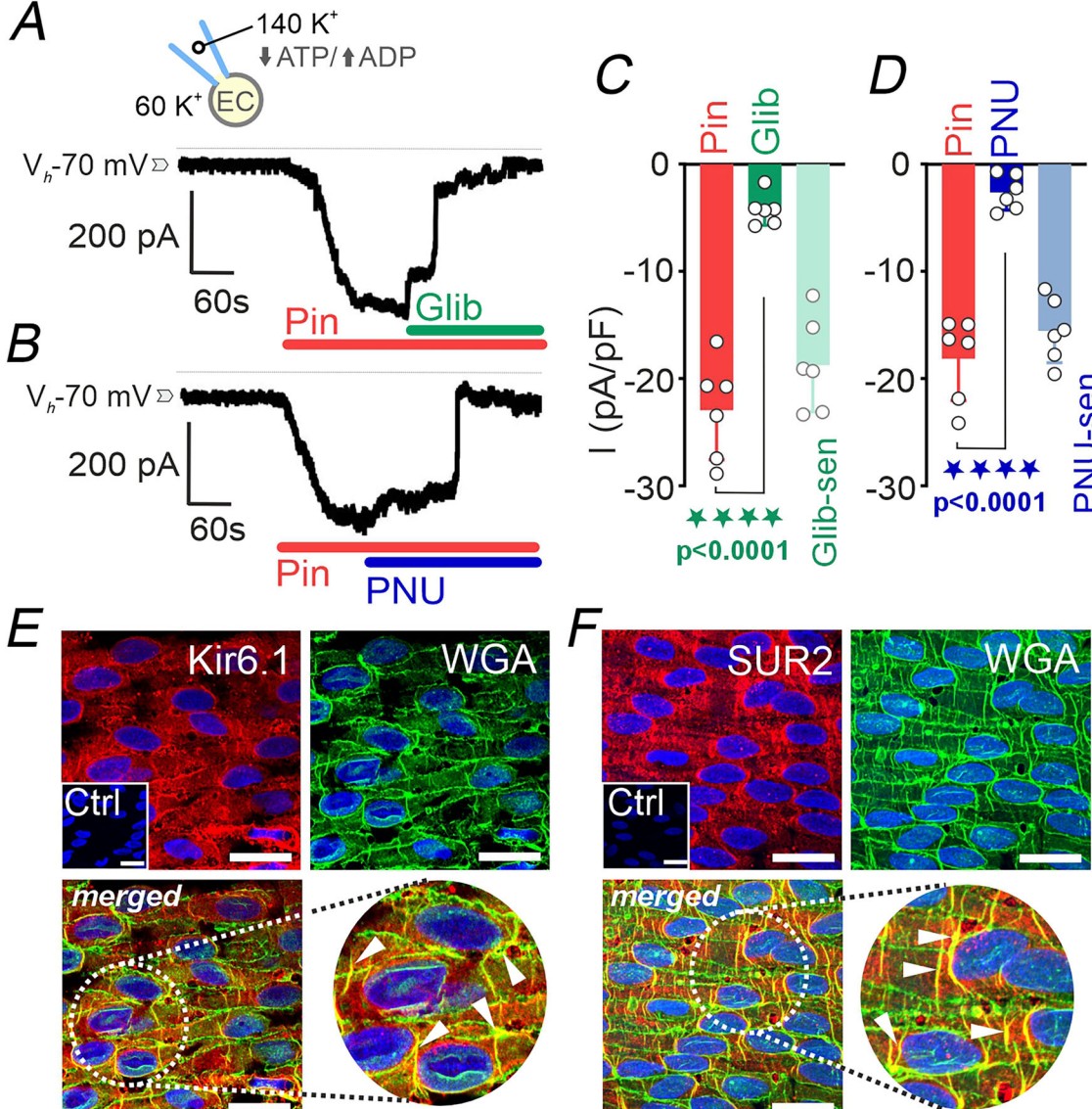

**Figure 5. Functional and molecular expression of K$_{ATP}$ channels in human pulmonary artery endothelial cells (PAECs)**

*A* and *B*, representative whole-cell current recordings from freshly isolated PAECs voltage-clamped at $-70$ mV, bathed in high-K$^+$ (60 mM) extracellular solution and dialysed with an internal solution containing low ATP and elevated ADP levels (0.1 mM each). Application of pinacidil (10 µM) elicited robust inward K$^+$ currents that were subsequently inhibited by glibenclamide (10 µM, *A*) or PNU-37883A (10 µM, *B*). *C* and *D*, summary data of current density (pA/pF) in response to pinacidil and its inhibition by glibenclamide (*C*; $n = 6$ cells, from six patients) or PNU-37883A (*D*; $n = 6$ cells, from six patients). Data are presented as means (SD), paired Student's *t* test. *E*, *en face* immunofluorescence staining of the Kir6.1 subunit (red, *top left*) and the plasma membrane marker WGA (green, *top right*); merged image (bottom left) with a magnified inset (bottom right). Corresponding negative control (Ctrl) is shown as inset. Arrowheads indicate examples of marker colocalization. *F*, equivalent immunostaining for the SUR2 subunit. Nuclei were counterstained with DAPI (blue) in all images. Each experiment was performed using cells from four independent patients; images are representative of 15–20 cells per patient. Scale bars = 20 µm.

We also explored the presence and functional significance of ATP-sensitive K⁺ channels in freshly isolated human PA SMCs and ECs. These channels are important regulators of membrane potential, typically coupling metabolic status to membrane excitability rather than being tonically active under basal conditions in a variety of vascular cells (Quayle et al., 1997; Sancho et al., 2022; Standen et al., 1989). Pharmacologically, K_{ATP} channels are activated by synthetic openers such as pinacidil and cromakalim and inhibited by oral hypo-glycaemic sulfonylurea drugs including glibenclamide and tolbutamide (Quayle et al., 1997; Sancho et al., 2022; Standen et al., 1989). While K_{ATP} currents had been reported in cultured pulmonary vascular cells from rats, bovines and humans (Chatterjee et al., 2003; Cui et al., 2002), and a recent study demonstrated the expression of Kir6.1, SUR2A and SUR2B subunits in human lung slices (Le Ribeuz et al., 2023), to our knowledge no study has directly examined K_{ATP} currents in freshly isolated human pulmonary vascular cells. Our findings demonstrate that both PAECs and PASMCs display pinacidil-activated, glibenclamide-sensitive K⁺ currents with the expected biophysical and pharmacological features of native K_{ATP} channels (Quayle et al., 1997; Sancho et al., 2022). These currents were also sensitive to PNU-37883A, which selectively blocks vascular-type K_{ATP} channels composed of Kir6.1 and SUR2B subunits (Li et al., 2013; Quayle et al., 1994; Sancho et al., 2022), a molecular identity further supported by immunohistochemistry. This notion is reinforced by single-cell RNA sequencing data for mouse lung vasculature, which reveal expression of *Kcnj8* and *Abcc9*, the genes encoding Kir6.1 and SUR2B, respectively, in both PASMCs and PAECs (He et al., 2018). Although interspecies differences must be considered, mRNA for Kir6.1 and SUR2B subunits has also been detected in cultured human PASMCs (Cui et al., 2002).

Functionally, K_{ATP} channels act as metabolic sensors – under conditions of low intracellular ATP or increased ADP, they open and induce membrane hyperpolarization, thereby limiting Ca²⁺ influx via voltage-gated Ca²⁺

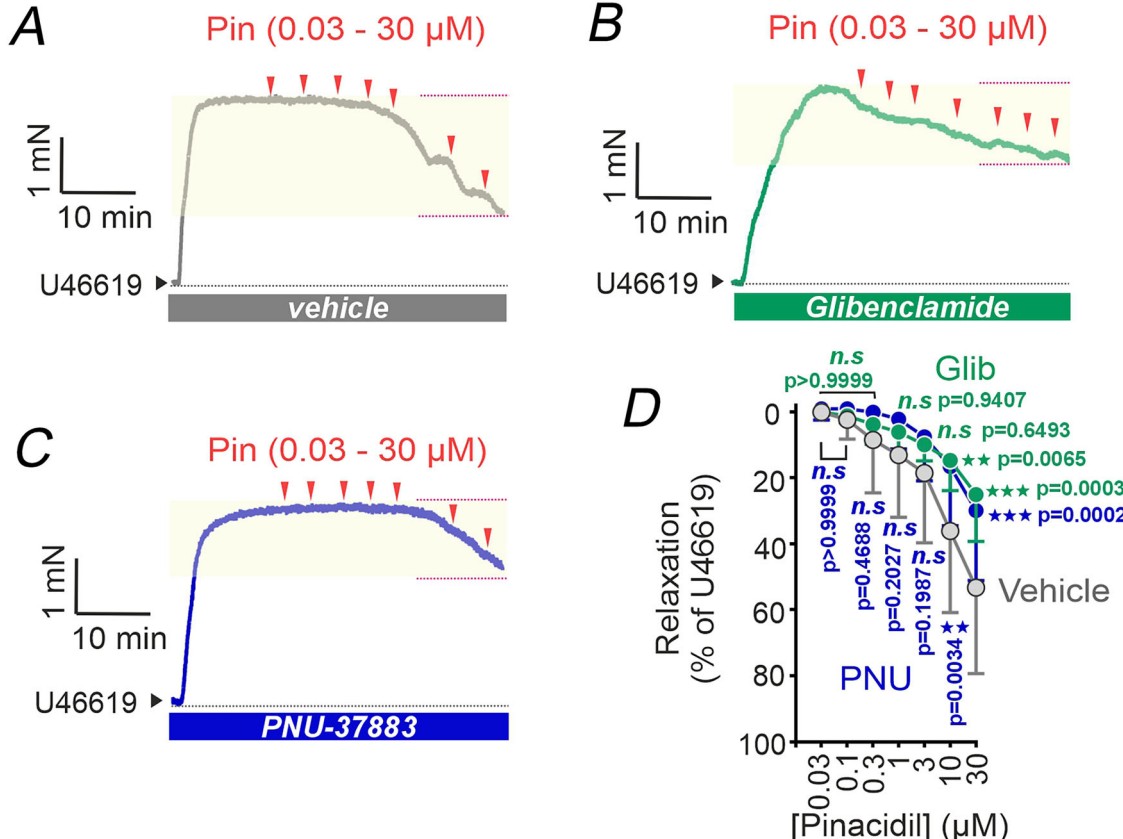

**Figure 6. K_{ATP} channel activation induces vasodilatation in human pulmonary arteries**
*A–C*, representative concentration–response curves showing pinacidil-induced relaxation of human PA rings precontracted with U46619 (0.1 μM), under vehicle-treated conditions (*A*), or following pretreatment with glibenclamide (10 μM) (*B*) or PNU-37883A (30 μM) (*C*). *D*, summary data showing concentration-dependent relaxation induced by pinacidil (0.03–30 μM) under vehicle-treated conditions and after glibenclamide or PNU-37883A. Data are presented as means (SD) (*n* = 7–12 PA rings from nine patients). *P* values indicate comparisons *versus* vehicle (two-way ANOVA followed by Bonferroni's multiple comparisons test).

channels, and promoting vasodilatation. Prior studies in rodent PASMCs (Clapp & Gurney, 1992; Smirnov et al., 1994), and intact vessels (Norton & Segal, 2018), have shown that $K_{ATP}$ channel activation leads to hyperpolarization, whereas inhibition causes depolarization, highlighting their central role in setting $V_M$ and modulating vascular tone. In human pulmonary arteries, Le Ribeuz et al. (2023) reported that pinacidil-induced activation of $K_{ATP}$ channels produces dose-dependent relaxation in vessels precontracted with U46619. Consistent with this, we found that pinacidil relaxed human PA rings under the same precontractile conditions to an extent similar to that previously reported (Wanstall et al., 1997). This effect was significantly attenuated by glibenclamide and PNU-37883A, providing direct functional evidence that Kir6.1/SUR2B $K_{ATP}$ channels contribute to the regulation of human pulmonary vascular tone.

$K_{ATP}$ channels have been extensively studied in systemic vascular beds, including the coronary and cerebral circulation (Quayle et al., 1997; Sancho et al., 2022; Zhao et al., 2020). However, their role in the pulmonary circulation remains less defined. Previous studies suggest that they modulate PA pressure during hypoxia or metabolic stress, but species- and model-specific differences have complicated interpretation of these findings (López-Valverde et al., 2005; Robertson et al., 1992). Intriguingly, gain-of-function mutations in the genes *KCNJ8* and *ABCC9* have been identified in Cantu syndrome, a rare genetic disorder in which PAH is frequently observed (Park et al., 2014). This association presents a paradox: how could enhanced $K_{ATP}$ channel activity, which typically promotes vasodilatation, contribute to a condition characterized by persistent vasoconstriction? Addressing this apparent contradiction requires direct measurements of $K_{ATP}$ channel activity in both PAECs and PASMCs under PAH conditions, which may reveal new insights into disease mechanisms and therapeutic targets for this devastating disease.

One notable limitation of the present study is that all experiments were conducted using lung tissue obtained from patients undergoing lobectomy for carcinoma. Although only macroscopically normal, non-tumorous sections of the PA were selected for analysis, we cannot fully exclude the possibility that underlying pathology, surgical stress, concurrent pharmacological treatments or systemic inflammatory responses may have subtly affected ion channel expression or vascular function.

Collectively, our results provide compelling evidence for the functional expression of Kir2 and $K_{ATP}$ channels in native human pulmonary arterial SMCs and ECs. Their presence in both vascular layers supports complementary roles in setting $V_M$ and regulating vascular tone. While Kir2 channels stabilize $V_M$ and amplify hyperpolarizing stimuli, $K_{ATP}$ channels act as metabolic sensors, linking energy status to excitability. This electro-metabolic coupling may be of particular relevance in the pulmonary circulation, where oxygenation, pressure, and metabolic demands fluctuate constantly. Our study raises the exciting possibility that dysregulation of Kir2 or $K_{ATP}$ channel activity contributes to pathological states such as PAH, characterized by disrupted vascular tone and endothelial dysfunction. Targeting these specific potassium channel subfamilies with selective modulators may offer a promising therapeutic strategy to counteract vascular impairment in PAH.

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

## Additional information

### Data availability statement

The data that support the findings of this study are available from the corresponding author upon reasonable request.

### Competing interests

The authors declare no conflicts of interest with respect to research, authorship and/or publication of this article.

### Author contributions

Conceptualization: M.S., F.P.-V.; Methodology: B.B., M.S., D.M.-C., L.M., R.A.; Formal analysis: B.B., M.S.; Investigation: B.B., M.S.; Resources: B.D.O.; Data curation: B.B., M.S.; Writing – original draft: M.S., B.B.; Writing – review and editing: B.B., M.S., D.M.-C., R.A., A.C., L.M., F.P.-V.; Project administration: B.B.; Funding acquisition: M.S. All authors have approved the final version of the manuscript and agree to be accountable for all

aspects of the work. All persons designated as authors qualify for authorship, and all those who qualify for authorship are listed.

## Funding

This work was financially supported by operating grants from the Fundación Contra la Hipertensión Pulmonar (FD10/21_01 to M.S.), Sociedad Española de Neumología y Cirugía Torácica SEPAR (1204-2022 to F.P.V.) and the Spanish Ministry of Science and Innovation (PID2023-147925OA-I00 to M.S., PID2020-117939RB-I00 to A.C., PID PID2019-107363RB-I00 to F.P.V.). RA was funded by EU Horizon 2020 Marie Skłodowska-Curie grant (No. 847635).

## Acknowledgements

The authors thank L.M. Alonso and M. Benito (Microscopy and Cytometry Centre at Complutense University of Madrid, Spain) for their technical assistance with the confocal microscopy and A. Ferruelo, R. Herrero and J.A. Lorente (Hospital Universitario de Getafe, Madrid, Spain) for providing access to the lung samples.

## Keywords

human smooth muscle and endothelial cells, Kir2 and K$_{ATP}$ channels, potassium channels, pulmonary artery, vascular tone

## Supporting information

Additional supporting information can be found online in the Supporting Information section at the end of the HTML view of the article. Supporting information files available:

**Peer Review History**

