## [Peer Review History · The Journal of Physiology]

Functional Expression of Inwardly Rectifying and ATP-sensitive Potassium Channels in Human Pulmonary Artery Smooth Muscle and Endothelial Cells

Bianca Barreira, Daniel Morales-Cano, Laura Moreno, Beatriz de Olaiz, Rui Adão, Angel Cogolludo, Francisco Perez-Vizcaino, and Maria Sancho

DOI: 10.1113/JP289445

Corresponding author(s): Maria Sancho (masanc75@ucm.es)

The following individual(s) involved in review of this submission have agreed to reveal their identity: Thomas Andrew Jepps (Referee #1)

Review Timeline:

Submission Date:	10-Jun-2025
Editorial Decision:	05-Aug-2025
Revision Received:	10-Dec-2025
Accepted:	06-Jan-2026

Senior Editor: Kim Barrett

Reviewing Editor: Nikki Jernigan

Transaction Report:

Dear Dr Sancho,

Re: JP-RP-2025-289445 "**Functional Expression of Inwardly Rectifying and ATP-sensitive Potassium Channels in Human Pulmonary Artery Smooth Muscle and Endothelial Cells**" by Bianca Barreira, Daniel Morales-Cano, Laura Moreno, Beatriz de Olaiz, Rui Adão, Angel Cogolludo, Francisco Perez-Vizcaino, and Maria Sancho

Thank you for submitting your manuscript to The Journal of Physiology. It has been assessed by a Reviewing Editor and by 2 expert referees and we are pleased to tell you that it is potentially acceptable for publication following satisfactory major revision.

REVISION CHECKLIST:

We look forward to receiving your revised submission.

Yours sincerely,

Kim Barrett
Senior Editor
The Journal of Physiology

REQUIRED ITEMS

1) - Author photo and profile. First or joint first authors are asked to provide a short biography (no more than 100 words for one author or 150 words in total for joint first authors) and a portrait photograph. These should be uploaded and clearly labelled together in a Word document with the revised version of the manuscript. See Information for Authors for further details.

2) - Please upload separate high-quality figure files via the submission form.

3) - Please ensure that any tables are editable and in Word format, and wherever possible, embedded in the article file itself.

4) - Please ensure that the Article File you upload is a Word file.

5) - Papers must comply with the Statistics Policy: https://jp.msubmit.net/cgi-bin/main.plex?form_type=display_requirements#statistics.

In summary:

- If $n \leq 30$, all data points must be plotted in the figure in a way that reveals their range and distribution. A bar graph with data points overlaid, a box and whisker plot or a violin plot (preferably with data points included) are acceptable formats.
- If $n > 30$, then the entire raw dataset must be made available either as supporting information, or hosted on a not-for-profit repository, e.g. FigShare, with access details provided in the manuscript.
- 'n' clearly defined (e.g. x cells from y slices in z animals) in the Methods. Authors should be mindful of pseudoreplication.
- All relevant 'n' values must be clearly stated in the main text, figures and tables.
- The most appropriate summary statistic (e.g. mean or median and standard deviation) must be used. Standard Error of the Mean (SEM) alone is not permitted.
- Exact p values must be stated. Authors must not use 'greater than' or 'less than'. Exact p values must be stated to three significant figures even when 'no statistical significance' is claimed.

6) - Please include an Abstract Figure file, as well as the Figure Legend text within the main article file. The Abstract Figure is a piece of artwork designed to give readers an immediate understanding of the research and should summarise the main conclusions. If possible, the image should be easily 'readable' from left to right or top to bottom. It should show the physiological relevance of the manuscript so readers can assess the importance and content of its findings. Abstract Figures should not merely recapitulate other figures in the manuscript. Please try to keep the diagram as simple as possible and

without superfluous information that may distract from the main conclusion(s). Abstract Figures must be provided by authors no later than the revised manuscript stage and should be uploaded as a separate file during online submission labelled as File Type 'Abstract Figure'. Please also ensure that you include the figure legend in the main article file. All Abstract Figures should be created using BioRender. Authors should use The Journal's premium BioRender account to export high-resolution images. Details on how to use and access the premium account are included as part of this email.

7)- Please ensure that all figures and tables have a title and legend, and that they have been cited within the main article text.

EDITOR COMMENTS

Reviewing Editor:

Methods Details: Reviewers have pointed out a number of missing details in the methods that should be addressed including whether sex differences were assessed, patient information, how vessel i.d. and viability were tested, software used for power analysis, and pressure used during experiments.

Data needs to be expressed using standard deviation, taking out +/- SEM, p-values need to be added instead of symbols.

The use of human vessels instead of cell culture is commendable and highly relevant. While this manuscript is interesting and has the potential for high impact, there are additional experiments that are needed to support your findings as outlined by the reviewers including verifying the contribution of Kir to resting membrane potential without the use of artificial conditions, clearly determining the functional smooth muscle vs endothelial contribution of K channels to contraction/relaxation, membrane and cellular localization of the channels using markers (CD31, SMA, plasma membrane) and cell surface biotinylation, and other minor points of clarification.

Please acknowledge prior studies on Katp channels (<https://doi.org/10.3389/fcvm.2022.1066047>)

In addition, please see the guidelines for statistical reporting. https://jp.msubmit.net/cgi-bin/main.plex?form_type=display_requirements#statistics

Data summaries should be presented as mean (SD). Please remove +/- and put the SD in parentheses. For a given conclusion to be assessed, the exact p values must be stated to three significant figures (not decimal places) even when 'no statistical significance' is being reported (i.e. for anything >0.001, please report to 3 significant figures, e.g. 0.00236 or 0.523, etc.). Asterisks alone should not be used to denote significance within figures.

Senior Editor:

Please see comments from Reviewing Editor

REFEREE COMMENTS

Referee #1:

The study by Barreira and others in the Sancho group investigates the functional role of Kir and KATP channels in isolated cells and intact human pulmonary arteries. The study is straightforward and well-written. The experiments presented are well performed and the data convincing. However, there are some key points that have not been addressed by this study, which reduces the overall impact of the presented findings. These concerns are listed below as suggestions for the authors to consider.

1) Although the myography experiments show that BaCl₂ can elicit a contraction of human pulmonary arteries (fig 3), the extent of this contraction is not clear. How does this contraction compared to a fully constricted arterial segment (e.g. effect of high ext[K⁺]? Can this contraction be inhibited by nifedipine or another L-type calcium channel blocker? This also raises a key point with regards to the overall contribution of Kir channels to the resting V_m. The actual contribution of Kir to V_m is not properly elucidated in the electrophysiological experiments since the conditions used enhance the contribution of Kir (and KATP) channels. Using high ext[K⁺] in these experiments, makes it difficult to conclude on the actual contribution of the Kir channels to resting V_m. Given the aim of the study and the focus of the discussion, this is a distinct weakness of the study. I would suggest performing current clamp experiments using a perforated patch on PSMCs to determine the effect of BaCl₂ (and glibenclamide) on the V_m.

2) Staying with figure 3, the relaxation of acetylcholine is rather small in comparison to what one would expect in any other endothelium-intact arteries. Can the authors offer an explanation for this small relaxation. Furthermore, it is not clear what role the endothelial Kir₂ channels have in the endothelial cells. The authors cannot conclude whether the BaCl₂ effect on the

acetylcholine relaxation is derived from the blockade of PASM or PAEC Kir channels. It would be useful to determine the effect of BaCl₂ on the sodium nitroprusside relaxation to at least show to what degree the effect seen in figure 3F is endothelium vs SMC derived. These experiments will also tell the authors whether the limited effect of acetylcholine is due to a lack of NO release, or is the maximum that one could expect the arterial segments to relax to NO.

3) Given the findings in figure 5, it is somewhat surprising that the authors did not investigate the effect of glibenclamide on the acetylcholine relaxation. These experiments are recommended, again to determine the role of endothelial KATP channels on vascular tone. The authors could also consider investigating the impact of pinacidil in the absence and presence of L-NAME and/or a TRAM/Apamin cocktail.

4) Given the points concerning endothelial integrity, was there a cut off used to demonstrate that the endothelium was intact in these experiments and could therefore be used to assess the role of BaCl₂ on an endothelial-dependent relaxation?

5) In the myography experiment, the authors state in the methods that they used a pressure of 30mmHg to determine the internal diameter. But what pressure were the segments set to at rest at the end of the normalisation procedure?

6) Were any sex-differences investigated? Were any of the patients on medication? Were any of the patients smokers?

7) In the final Key Point, the authors claim that these channels could contribute to PAH. Although this is fine to speculate on in the discussion, there is no evidence of this in the current study, therefore I suggest this is removed as a key point.

Referee #2:

The authors addressed the expression of two potassium channels (Kir2 and KATP channels) in native human pulmonary artery smooth muscle and endothelial cells. By using pharmacological compounds, they also studied the role of these both channels in the pulmonary arterial vasoreactivity (contraction and relaxation).

Channel expression is known to be altered in cultured pulmonary arterial cells, therefore importantly the present study is performed in native and not cultured cells and this is a strong point of the study.

The results are clearly presented and convincing however some concerns are still present in the manuscript.

Major comments:

First of all, the authors claim that their findings demonstrate for the first time the functional presence of Kir2 and KATP channel in native human pulmonary arterial cells but this is partially true. Indeed, the presence of KATP has already been shown by Le Ribeuz et al. (Front Cardiovasc Med, 2023) in human lung slices from control patients. They show that Kir6.1, SUR2A and SUR2B are indeed expressed in both smooth muscle and endothelium. Moreover, like in the present study, Le Ribeuz and her collaborators have also shown that pinacidil (KATP activator) dose-dependently induced a relaxation in human pulmonary arteries precontracted with U46619. Such results are not even discussed in the present manuscript. Although the physiological interest of the present study is high, the novelty is therefore less due to the previous study of Le Ribeuz et al.

Specific comments:

Methods:

1- The authors should precise how they are sure that the vessels they isolate are human resistance arteries and not veins or bronchial vessels.

2- Is the contraction to 80 mM potassium used to test the viability of the vessels? If yes, do the authors exclude vessels with a too small contraction to potassium and what is the value used for exclusion in that case?

3- What is the software and the protocol used for power analysis?

Result section

4- In all the manuscript the authors claim that the channels are expressed on the membrane but this is not always convincing (figure 1F, G, J, K, figure 2E, H, G, J, figure 4F, G, J, K, figure 5E, G). A double labelling (channel and membrane marker) should be addressed to be sure that the channels are localized at the membrane. Moreover, cell surface biotinylation followed by western blotting could be performed to quantify the number of channels localized at the membrane.

5- Figure 2, the immunofluorescent staining has been observed on the endothelial face of the vessel by using confocal microscopy. However, since there is only one layer of endothelial cells, how can the authors be sure that they observe the endothelium and not the smooth muscle. Nuclei for endothelial cells are usually rounder than the nuclei of smooth muscle cells and figure 2F and 2I show nuclei that look like nuclei of smooth muscle. Double labelling of the endothelium with CD31

or vWF for instance and the potassium channels could be performed.

6- Figure 3, the delta value (in mN/mm²) of the contraction to high potassium should be given for comparison with the contraction to BaCl₂.

7- It would help to number the pages of the manuscript.

END OF COMMENTS

EDITOR COMMENTS

Reviewing Editor:

Methods Details: Reviewers have pointed out a number of missing details in the methods that should be addressed including whether sex differences were assessed, patient information, how vessel i.d. and viability were tested, software used for power analysis, and pressure used during experiments.

Data needs to be expressed using standard deviation, taking out +/- SEM, p-values need to be added instead of symbols.

The use of human vessels instead of cell culture is commendable and highly relevant. While this manuscript is interesting and has the potential for high impact, there are additional experiments that are needed to support your findings as outlined by the reviewers including verifying the contribution of Kir to resting membrane potential without the use of artificial conditions, clearly determining the functional smooth muscle vs endothelial contribution of K channels to contraction/relaxation, membrane and cellular localization of the channels using markers (CD31, SMA, plasma membrane) and cell surface biotinylation, and other minor points of clarification.

Please acknowledge prior studies on Katp channels
(<https://doi.org/10.3389/fcvm.2022.1066047>)

In addition, please see the guidelines for statistical reporting. https://jp.msubmit.net/cgi-bin/main.plex?form_type=display_requirements#statistics

Data summaries should be presented as mean (SD). Please remove +/- and put the SD in parentheses. For a given conclusion to be assessed, the exact p values must be stated to three significant figures (not decimal places) even when 'no statistical significance' is being reported (i.e. for anything >0.001, please report to 3 significant figures, e.g. 0.00236 or 0.523, etc.). Asterisks alone should not be used to denote significance within figures.

Senior Editor:

Please see comments from Reviewing Editor

Response: We thank the Reviewing Editor and Senior Editor for their careful evaluation of our manuscript and for providing constructive guidance to improve its clarity, rigor, and completeness. We have carefully addressed all points raised, including:

1. **Methods details:** We have now included comprehensive information on patient characteristics, sex distribution, vessel internal diameter and viability assessment, pressure used during experiments and other relevant aspects. These updates are detailed in the Methods section.
2. **Statistical reporting:** All data are now presented as mean (SD), and exact p-values are reported to three significant figures throughout, in accordance with the journal's guidelines. Figure panels have been updated accordingly.

3. **Channel localization and functional contributions:** We performed additional experiments to clarify the cellular localization of Kir2 and K_{ATP} channel subunits, including double labelling with the plasma membrane marker WGA. We also provide detailed descriptions of the *en face* endothelial preparation and cell type identification using CD31 (endothelium) or α -smooth muscle actin (smooth muscle). Furthermore, electrophysiology recordings were performed to assess the contribution of Kir2 channels to resting membrane potential in pulmonary artery (PA) smooth muscle cells. Vascular reactivity experiments were also performed as suggested by the Referees. These new data have been incorporated in the revised manuscripts and corresponding Figures.
4. **Acknowledgement of prior studies:** We have now cited previous work on K_{ATP} channels, including the suggested reference, highlighting how our study complements and extends these findings.

REFEREE COMMENTS

Referee #1:

The study by Barreira and others in the Sancho group investigates the functional role of Kir and K_{ATP} channels in isolated cells and intact human pulmonary arteries. The study is straightforward and well-written. The experiments presented are well performed and the data convincing. However, there are some key points that have not been addressed by this study, which reduces the overall impact of the presented findings. These concerns are listed below as suggestions for the authors to consider.

Response: We thank Referee 1 for taking the time to carefully review our manuscript and provide constructive comments.

1) Although the myography experiments show that $BaCl_2$ can elicit a contraction of human pulmonary arteries (fig 3), the extent of this contraction is not clear. How does this contraction compared to a fully constricted arterial segment (e.g. effect of high $ext[K^+]$)? Can this contraction be inhibited by nifedipine or another L-type calcium channel blocker? This also raises a key point with regards to the overall contribution of Kir channels to the resting V_m . The actual contribution of Kir to V_m is not properly elucidated in the electrophysiological experiments since the conditions used enhance the contribution of Kir (and K_{ATP}) channels. Using high $ext[K^+]$ in these experiments, makes it difficult to conclude on the actual contribution of the Kir channels to resting V_m . Given the aim of the study and the focus of the discussion, this is a distinct weakness of the study. I would suggest performing current clamp experiments using a perforated patch on PSMCs to determine the effect of $BaCl_2$ (and glibenclamide) on the V_m .

Response: We thank the reviewer for their insightful comments. To clarify the magnitude of the $BaCl_2$ -induced contraction, we have now expressed this response as a percentage of the maximal contraction elicited by high extracellular K^+ (80 mM) (Figure 3C; Methods: page 7, lines 294–296). As shown in updated Fig. 3C, $BaCl_2$ (100 μ M) induces a partial contraction (~35% of the maximal KCl response), supporting a moderate but physiologically relevant tonic hyperpolarizing influence of Kir2 channels under basal conditions (Results, page 9, lines 400–402). To avoid potential bias, rings were randomly allocated to either time control

or Ba²⁺ treatment groups. The average KCl-induced contraction was comparable between the two groups: Control: 0.89 (0.55) mN/mm²; Ba²⁺-treated: 0.92 (0.40) mN/mm² (no significant differences; $p = 0.877$), confirming equivalent contractile capacity.

In response to the reviewer's question regarding the involvement of L-type Ca²⁺ channels, we performed additional myography experiments in the presence of nifedipine (1 μ M). As shown in updated Fig. 3B-C, nifedipine essentially blunted BaCl₂-induced contractions, demonstrating that Kir2 channel inhibition mediates contraction through membrane depolarization and subsequent activation of L-type Ca²⁺ channels. These findings have been included in the revised manuscript (Fig. 3B-C; Results, page 9, lines 402–408).

We agree that current-clamp recordings under near-physiological conditions represent the most direct approach to assess ΔV_M . Our initial electrophysiological experiments were designed to accurately identify Kir2 and K_{ATP} currents based on their pharmacology and I-V profiles, under conditions optimized for current resolution (i.e. high extracellular K⁺; low intracellular ATP for K_{ATP} recordings). These were complemented by intact-tissue myography to assess translational relevance for vascular tone under physiological conditions. Although this strategy does not allow precise quantification of the channel's absolute contribution to resting V_M, it confirms their functional presence and influence under resting conditions (Kir2) or following pharmacological modulation (K_{ATP}).

Considering the reviewer's suggestion, we have now performed additional current-clamp experiments on human pulmonary artery smooth muscle cells (PASMCs) using the conventional whole-cell configuration under physiological extracellular [K⁺]. Despite not preserving the intracellular milieu to the same extent as perforated-patch, BaCl₂ produced a robust depolarization (~10-15 mV), supporting a contribution of Kir2 channels to resting V_M. These new data have been incorporated into the revised manuscript (Results, page 8, lines 356–360) and presented in Fig. 1F-G.

Finally, with regard to K_{ATP} channels, we believe that testing glibenclamide would not provide meaningful information about their contribution to resting V_M, as these channels do not appear to be tonically active under basal conditions. This interpretation is supported by our myography experiments, in which glibenclamide (10 μ M) did not induce contraction of human pulmonary artery (PA) rings. Accordingly, we have clarified in the revised manuscript that K_{ATP} channels primarily modulate V_M in response to metabolic changes or pharmacological activation rather than under resting conditions.

2) Staying with figure 3, the relaxation of acetylcholine is rather small in comparison to what one would expect in any other endothelium-intact arteries. Can the authors offer an explanation for this small relaxation. Furthermore, it is not clear what role the endothelial Kir2

channels have in the endothelial cells. The authors cannot conclude whether the BaCl₂ effect on the acetylcholine relaxation is derived from the blockade of PASMC or PAEC Kir channels. It would be useful to determine the effect of BaCl₂ on the sodium nitroprusside relaxation to at least show to what degree the effect seen in figure 3F is endothelium vs SMC derived. These experiments will also tell the authors whether the limited effect of acetylcholine is due to a lack of NO release, or is the maximum that one could expect the arterial segments to relax to NO.

Response: We agree with the reviewer that the magnitude of ACh-induced relaxation observed in our human PA preparations (~43% at 100 μM ACh) appears modest when compared with typical values reported in endothelium-intact systemic arteries or rodent PAs. However, this range is consistent with previous findings in human PAs obtained from patients undergoing lung resection (Pandolfi et al., 2017; PMID: 27701117), where endothelial function is often compromised due to patient-related factors such as advanced age, smoking history, or coexisting cardiovascular disease. Supporting this interpretation, our laboratory routinely records larger ACh-induced relaxations (~60-70% at 100 μM ACh) in rodent PAs (Mondejar-Parreño et al., 2019; PMID: 31432395), confirming that the limited relaxation observed in the present study likely reflects the variable and reduced endothelial viability of human surgical specimens.

In our study, PA rings were pre-contracted using U46619, a thromboxane A₂ analogue that produces stable and reproducible contractions in human PAs (Cogolludo et al., 2005; PMID: 15769451; Moral-Sanz et al., 2011; PMID: 21490312). Although alternative agonists (e.g. serotonin, phenylephrine) produced inconsistent responses, a known limitation of U46619 is that it tends to generate contractions that are less sensitive to endothelium-dependent vasodilation. This further contributes to the modest ACh-induced relaxation observed. We have now acknowledged in the revised manuscript that this response was modest but comparable with previous studies in human pulmonary arteries (Results: page 10, lines 413–415)

To address the reviewer's concern regarding the relative contribution of endothelial vs. smooth muscle Kir2 channels, we conducted additional experiments evaluating the effect of BaCl₂ on sodium nitroprusside (SNP)-induced relaxation. Our results show that BaCl₂ did not significantly affect SNP responses, indicating that PASMC sensitivity to NO is preserved. The similar magnitude of relaxation to SNP in our preparations further suggests that the limited ACh-induced relaxation reflects the amount of NO released by the endothelium rather than a limitation of smooth muscle relaxability. Therefore, the inhibitory effect of Ba²⁺ on ACh-induced relaxation is most consistent with blockade of endothelial Kir2 channels. These new results have been incorporated into the revised manuscript (Figures 3G-H, Results, page 10, lines 417–423).

3) Given the findings in figure 5, it is somewhat surprising that the authors did not investigate the effect of glibenclamide on the acetylcholine relaxation. These experiments are recommended, again to determine the role of endothelial K_{ATP} channels on vascular tone.

Response: We thank the reviewer for the suggestion regarding the potential involvement of endothelial K_{ATP} channels in ACh-mediated relaxation. Endothelial K_{ATP} can regulate V_M in certain vascular beds in response to specific endogenous vasoactive stimuli via Gs-coupled receptors, such as adenosine or CGRP (Sancho et al., 2022; PMID: 35349300) or hypoxia,

and we plan to explore these pathways in future studies. However, under basal conditions and during ACh stimulation, endothelial K_{ATP} channels appear largely inactive, and pharmacological inhibition with glibenclamide is therefore unlikely to produce a measurable effect.

To confirm this, we performed concentration-response curves to ACh in the absence and presence of the K_{ATP} channel inhibitor PNU-37883A (30 μM) in human PA rings. As expected, PNU did not significantly alter the ACh-induced relaxation (please see graph below). These results support our idea that endothelial K_{ATP} channels are largely inactive under these conditions. Although these experiments could potentially be included as supplementary material, we opted not to present them in the manuscript in line with the journal's preference to minimize supplementary files, as they do not provide additional mechanistic insight beyond what is already reported.

Regarding the reviewer's suggestion to test pinacidil in the absence or presence of L-NAME (100 μM) and/or a TRAM-34 (1 μM)/Apamin (100 nM) cocktail, we have performed these experiments in human PA rings. The presence of these blockers did not significantly alter pinacidil-induced relaxation, indicating that under our experimental conditions, the nitric oxide (NO) and the endothelium-derived hyperpolarizing factors pathways does not contribute appreciably to the response. Although these findings confirm the lack of involvement of these pathways, we have opted not to include the data in the manuscript, as they fall outside the primary scope of the present study and do not provide additional mechanistic insight relevant to our main conclusions.

4) Given the points concerning endothelial integrity, was there a cut off used to demonstrate that the endothelium was intact in these experiments and could therefore be used to assess the role of BaCl₂ on an endothelial-dependent relaxation?

Response: We agree with the reviewer that a clear definition of endothelial integrity is essential for the interpretation of endothelium-dependent relaxations. In our study, only PA rings exhibiting $\geq 20\%$ relaxation to 100 μM ACh following pre-constriction to U46619 were considered endothelium-intact and included in experiments assessing endothelial mechanisms. Rings showing $< 20\%$ relaxation to ACh were classified as endothelium-compromised and excluded from these analyses. These predefined inclusion criteria have now been explicitly added to the Methods section (page 7; lines 300–301).

5) In the myography experiment, the authors state in the methods that they used a pressure of 30mmHg to determine the internal diameter. But what pressure were the segments set to at rest at the end of the normalisation procedure?

Response: We thank the reviewer for highlighting this missing detail. We have clarified in the Methods section (page 7, lines 280–288) that, for each artery, the passive wall-tension-internal circumference relationship was determined, and vessels were normalized to an internal circumference corresponding to $0.9 \times L_{30}$. Following normalization, arterial segments were maintained at a transmural pressure equivalent to 30 mmHg during the equilibration period prior to functional assessment. This pressure was chosen based on previous studies in pulmonary arteries (Ozaki et al., 1998; PMID: 9843843) and reflects the physiological pressure experienced by these vessels *in situ*. Maintaining this pressure enables an accurate assessment of basal tone and contractile responses under near-physiological conditions.

6) Were any sex-differences investigated? Were any of the patients on medication? Were any of the patients smokers?

Response Regarding sex differences, the limited number of male and female patients in our cohort precludes robust statistical analysis, and therefore, sex-specific differences were not specifically investigated in the present study. Patient clinical information, including medication use and smoking history, was obtained from surgical records. A subset of patients was taking cardiovascular or respiratory medications, and several had a history of smoking. These factors are inherent to studies using human surgical specimens and are acknowledged as potential sources of variability in vascular responses. We have added a statement in the Methods section to clarify patient characteristics and their potential influence on ion channel currents or vascular responses (page 4; lines 170–175).

7) In the final Key Point, the authors claim that these channels could contribute to PAH. Although this is fine to speculate on in the discussion, there is no evidence of this in the current study, therefore I suggest this is removed as a key point.

Response: We agree that the potential contribution of Kir2 and K_{ATP} channels to pulmonary arterial hypertension (PAH) is speculative and not directly addressed by the present study. Accordingly, we have removed this statement from the Key Points. The possible relevance of

Kir2 and K_{ATP} dysregulation to PAH is still discussed in the Discussion section, where it is clearly framed as a hypothesis to be investigated in future studies.

Referee #2:

The authors addressed the expression of two potassium channels (Kir2 and K_{ATP} channels) in native human pulmonary artery smooth muscle and endothelial cells. By using pharmacological compounds, they also studied the role of these both channels in the pulmonary arterial vasoreactivity (contraction and relaxation).

Channel expression is known to be altered in cultured pulmonary arterial cells, therefore importantly the present study is performed in native and not cultured cells and this is a strong point of the study.

The results are clearly presented and convincing however some concerns are still present in the manuscript.

Response: We thank Referee 2 for their positive evaluation of our work and for the constructive comments provided, which helped us further improve the manuscript.

Major comments:

First of all, the authors claim that their findings demonstrate for the first time the functional presence of Kir2 and K_{ATP} channel in native human pulmonary arterial cells but this is partially true. Indeed, the presence of K_{ATP} has already been shown by Le Ribeuz et al. (Front Cardiovasc Med, 2023) in human lung slices from control patients. They show that Kir6.1, SUR2A and SUR2B are indeed expressed in both smooth muscle and endothelium. Moreover, like in the present study, Le Ribeuz and her collaborators have also shown that pinacidil (K_{ATP} activator) dose-dependently induced a relaxation in human pulmonary arteries precontracted with U46619. Such results are not even discussed in the present manuscript. Although the physiological interest of the present study is high, the novelty is therefore less due to the previous study of Le Ribeuz et al.

Response: We thank the reviewer for highlighting the study by Le Ribeuz et al. (2023). We fully acknowledge that their work demonstrated the expression of K_{ATP} channel subunits (Kir6.1, SUR2A, SUR2B) in human pulmonary artery smooth muscle and endothelium using paraffin-embedded lung sections, and that activation of these channels by pinacidil induces relaxation in U46619-precontracted arteries.

Our study complements and extends these findings by providing the first direct electrophysiological characterization of Kir2 channels alongside K_{ATP} channels in freshly isolated human pulmonary artery smooth muscle and endothelial cells. By combining patch-clamp electrophysiology and wire myography, we link ion channel activity to membrane potential and vascular tone and quantify the relative contribution of these channels to basal tone and endothelium-dependent relaxation. We believe that this mechanistic insight represents the novel aspect of our work.

We have now referenced the study by Le Ribeuz et al. (2023) in the Discussion section, emphasizing the complementary nature of our findings and clarifying the specific novelty of the present study (page 13, lines 574–577; page 13, lines 595–597).

Specific comments:

Methods:

1- The authors should precise how they are sure that the vessels they isolate are human resistance arteries and not veins or bronchial vessels.

Response: Pulmonary arteries were carefully dissected from peripheral lung parenchyma under a stereomicroscope. Only vessels displaying a typical arterial morphology (thicker wall, round cross-sectional profile, and clearly defined smooth muscle layers) were selected, while veins (thinner walls, irregular shape, no clear smooth muscle layers) and bronchial structures were excluded. These methodological details have now been clarified in the Methods section (page 4; lines 184–188). Importantly, bronchial rings respond to acetylcholine with contraction rather than relaxation (Rhoden et al., 1988; PMID: 2844715), providing a clear functional distinction from pulmonary arteries and further validating the arterial identity of the isolated vessels.

2- Is the contraction to 80 mM potassium used to test the viability of the vessels? If yes, do the authors exclude vessels with a too small contraction to potassium and what is the value used for exclusion in that case?

Response: Yes, the contraction induced by 80 mM KCl was used to assess the functional viability of the pulmonary artery rings. As now clarified in the revised manuscript (Methods: page 7, lines 288–292), arteries that did not develop a measurable contraction in response to this KCl challenge were considered non-viable and excluded from further experiments. Specifically, vessels failing to achieve a minimum developed tension of 0.25 mN/mm² were excluded.

3- What is the software and the protocol used for power analysis?

Response: Power analysis was performed using G*Power 3.1.9.7 software (Faul et al., 2007; PMID: 17695343). Calculations were based on previously published data and pilot results from our laboratory, assuming a statistical power of 80% ($\beta=0.20$) and a significance level of $\alpha=0.05$. For vascular reactivity experiments, sample size estimates were based on the expected differences in maximal relaxation between groups. For patch-clamp recordings, the effect size was derived from the anticipated changes in K⁺ current amplitude or V_M under pharmacological conditions. Because human pulmonary tissue is limited and the yield of viable arterial rings or isolated cells varies between donors, not all experiments resulted in identical sample sizes. Nonetheless, the final n values fall within the range commonly used in comparable published studies. We have now clarified this in the revised manuscript (Methods: page 8, lines 324-329).

Result section

4- In all the manuscript the authors claim that the channels are expressed on the membrane but this is not always convincing (figure 1F, G, J, K, figure 2E, H, G, J, figure 4F, G, J, K, figure 5E, G). A double labelling (channel and membrane marker) should be addressed to be sure that the channels are localized at the membrane. Moreover, cell surface biotinylation

followed by western blotting could be performed to quantify the number of channels localized at the membrane.

Response: We thank the reviewer for this important and constructive comment. To address this concern, we performed double immunofluorescence labeling of the channel subunits of interest together with wheat germ agglutinin (WGA), a well-established plasma membrane marker, in both isolated PASMCs and *en face* endothelial preparations. As shown in the revised Figures 1I-J; 2F-G; 4F-G; 5E-F), we observed a strong colocalization signal between the studied subunits and WGA, supporting their presence at the plasma membrane with some residual staining in the cytosol. These new images have been incorporated into the corresponding Figures, and the Results section has been updated accordingly (Results: page 9, lines 360–364; page 9, lines 389–394; page 10, lines 446–449; page 11 lines 466–468).

In the present study, immunofluorescence was used primarily to complement the functional experiments, which provide direct and irrefutable evidence of Kir2 and K_{ATP} channel activity at the plasma membrane. While quantification of membrane-localized channels via biochemical approaches such as cell surface biotinylation followed by Western blotting would be of interest, this is beyond the scope of the current manuscript and technically challenging due to the limited availability of fresh human tissue. We acknowledge this limitation and consider it an important avenue for future investigation.

5- Figure 2, the immunofluorescent staining has been observed on the endothelial face of the vessel by using confocal microscopy. However, since there is only one layer of endothelial cells, how can the authors be sure that they observe the endothelium and not the smooth muscle. Nuclei for endothelial cells are usually rounder than the nuclei of smooth muscle cells and figure 2F and 2I show nuclei that look like nuclei of smooth muscle. Double labelling of the endothelium with CD31 or vWF for instance and the potassium channels could be performed.

Response: We appreciate the reviewer's comment. The *en face* preparations were performed by longitudinally opening the pulmonary arteries and pinning them on Sylgard-coated plates, thereby exposing the endothelial surface for immunofluorescence as previously described (Sancho et al., 2019 PMID: 31043073). The mounted preparations were imaged using confocal microscopy with z-stacks, as illustrated in the revised Figure 2E.

Under the microscope, the endothelial layer can be easily identified: PAECs nuclei are rounder and arranged with a slightly irregular pattern, followed by the internal elastic lamina (IEL), and then multiple layers of PASMCs with elongated nuclei oriented perpendicular to those of the endothelium. As suggested by the reviewer, the identity of the endothelium was further confirmed by positive CD31 staining. This information has been added to the revised manuscript (Results: page 9, lines 383–389).

We also acknowledge that some of the previously shown images in Figs. 2F and 2I may not have been fully stretched, which could have made it difficult to distinguish the endothelial layer. To address this, we have acquired new immunofluorescence images, now included in the revised Figures 2F-G and 5E-F.

Additionally, although not specifically requested by the reviewer, we also performed α -actin immunolabeling in freshly isolated PASMCs to further confirm their identity (Figure 1H, Results: page 9, lines 360–362).

6- Figure 3, the delta value (in mN/mm²) of the contraction to high potassium should be given for comparison with the contraction to BaCl₂.

Response: We thank the reviewer for this helpful suggestion. To enable direct comparison, the contraction induced by BaCl₂ is now expressed as a percentage of the response to high potassium. These details have been incorporated into the Figure 3C and clarified in the Methods section as: “...*contractile responses to BaCl₂ were expressed as a percentage of the maximal contraction elicited by high potassium (80 mM)*” (page 7, lines 295-296).

7- It would help to number the pages of the manuscript.

Response: We apologize for this oversight. Page numbers have now been added throughout the manuscript, and line numbers on each page have also been included to facilitate the review process.

Dear Dr Sancho,

Re: JP-RP-2025-289445R1 "**Functional Expression of Inwardly Rectifying and ATP-sensitive Potassium Channels in Human Pulmonary Artery Smooth Muscle and Endothelial Cells**" by Bianca Barreira, Daniel Morales-Cano, Laura Moreno, Beatriz de Olaiz, Rui Adão, Angel Cogolludo, Francisco Perez-Vizcaino, and Maria Sancho

We are pleased to tell you that your paper has been accepted for publication in The Journal of Physiology.

Yours sincerely,

Kim Barrett
Senior Editor
The Journal of Physiology

IMPORTANT POINTS TO NOTE FOLLOWING ACCEPTANCE OF YOUR PAPER:

- **IMPORTANT NOTICE ABOUT OPEN ACCESS:** To assist authors whose funding agencies mandate immediate public access to published research findings, The Journal of Physiology allows authors to pay an Open Access (OA) fee to have their papers made freely available immediately on publication.

- You can help your research get the attention it deserves! Check out Wiley's free Promotion Guide for best-practice recommendations for promoting your work at: www.wileyauthors.com/eoo/guide. You can learn more about Wiley Editing Services which offers professional video, design, and writing services to create shareable video abstracts, infographics, conference posters, lay summaries, and research news stories for your research at: www.wileyauthors.com/eoo/promotion.

- If you would like to receive our 'Research Roundup', a monthly newsletter highlighting the cutting-edge research published in The Physiological Society's family of journals (The Journal of Physiology, Experimental Physiology, Physiological Reports, The Journal of Nutritional Physiology and The Journal of Precision Medicine: Health and Disease), please click this link, fill in your name and email address and select 'Research Roundup': <https://www.physoc.org/journals-and-media/membernews>

EDITOR COMMENTS

Reviewing Editor:

Nice study on naive potassium channels in human pulmonary arteries, no further comments.

REFEREE COMMENTS

Referee #1:

The authors have addressed my comments. No further concerns.

Referee #2:

The revised manuscript has now been strongly improved. The immunofluorescent pictures are very convincing.

However, regarding comment 1 on methods, I agree that bronchial rings respond to acetylcholine with contraction rather than a relaxation but my comment was not on bronchial rings but on bronchial arterial rings.